# Comparative study of protease hydrolysis reaction demonstrating Normalized Peptide Bond Cleavage Frequency and Protease Substrate Broadness Index

**Shukun Yu**  *, **Janne Bech Thoegersen, Karsten Mathias Kragh**

DuPont Nutrition & Biosciences, Brabrand, Aarhus, Denmark

* shukun.yu@dupont.com

## Abstract

Two commercial proteases (subtilisin-typed FNA from *Bacillus amyloliquefaciens*, and chymotrypsin-like NPP from *Nocardiopsis prasina*), porcine pepsin, porcine pancreatin having protease activity and their combinations were studied *in vitro* by LC-MS for their ability to digest soy protein isolate (SPI) under conditions close to those found in the stomach (pH 3.7) and small intestine (pH 6.5). The total number of peptides generated, and their size distribution were obtained under each set of the digestion conditions. These peptides were grouped according to their C-terminal amino acid (AA) residue (P1) and mass, based on which two concepts were proposed, *i.e.*, Normalized Peptide Bond Cleavage Frequency (NPBCF) and Protease Substrate Broadness Index (PSBI). At pH 3.7, FNA+pepsin increased PSBI *vs*. pepsin alone by 2.7 and 4.9 percentage points (p.p.) at a SPI:protease ratio of 20:1 and 100:1, respectively. At pH 6.5, FNA+pancreatin improved PSBI by 9.1 and 10.2 p.p. at SPI:protease 20:1 and 100:1, respectively, *vs*. pancreatin alone. NPP generated 38% more peptides than FNA when administered with pancreatin at SPI:protease 200:1:1 and pH 6.5, but FNA alone (28.9) or FNA+pancreatin (29.1) gave a higher PSBI than pancreatin (22.2), NPP (20.3) and NPP+pancreatin (22.0). At pH 3.7 FNA generated 59% and 39% of peptides of pepsin at SPI:protease of 20:1 and 100:1, respectively, and both groups of peptides had similar size distribution. At pH 6.5 more small sized peptides were generated by FNA or FNA+pancreatin than pancreatin and NPP alone or pancreatin+NPP. In conclusion, FNA showed complementary effects with pepsin and pancreatin in terms of PSBI and generated more small sized peptides compared to NPP.

## Introduction

Protein is a major and expensive component of animal feed, accounting for about 20% of the total dry matter content of swine and poultry diets. Soybean meal is the leading source of feed protein [1]. In the gut, protein is hydrolyzed by endogenous proteases, principally pepsin (EC 3.4.23.1), trypsin (EC 3.4.21.4) and chymotrypsin (EC 3.4.21.1), releasing peptides that are

**Data Availability Statement:** All data are given under Support information as S1, S2 and S3 Tables.

**Funding:** E. I. du Pont de Nemours and Company provided support in the form of R&D budget for all authors. The specific roles of these authors are articulated in the 'author contributions' section. The funder had no role in study design, data collection and analysis, decision if it should be published or not and on how to publish, or preparation of the manuscript.

**Competing interests:** E. I. du Pont de Nemours and Company provided support in the form of R&D budget for all authors. This does not alter our adherence to PLOS ONE policies on sharing data and materials. The protease FNA and soy protein isolate Supro 760 described in the "M&M" section are commercial products provided by DuPont for this research. There are no patents, products in development or other marketed products associated with this research to be declared.

processed further by intestinal brush border anchored peptidases and amino acids (AAs) essential for growth and maintenance. However, digestion by endogenous protease is incomplete. Recent studies using proteomics approaches have revealed the presence of partially digested proteins from soybean meal (and other legume and cereal feed ingredients) in broiler digesta [2, 3]. It is estimated that only about 80% of dietary soybean meal is digested by swine and poultry [4, 5] or even less in high fiber, high phytate diets such as those containing industrial by-products [6,7]. Undigested protein is wasteful from an ingredient cost perspective and its excretion can result in the accumulation of harmful ammonia gas and environmental pollution due to the leeching of nitrogen from feces into the ground and water courses [8].

Exogenous proteases, usually from microbial (bacterial and fungal) sources, are increasingly being used in swine and poultry diets to increase feed protein hydrolysis. The science underpinning their precise mode of action and efficacy in different production and dietary settings is still developing. As a result, their use is not yet as widespread as phytase and xylanase. When administered alone (as a mono-component feed additive), it is clear that exogenous protease can improve apparent ileal AA digestibility, growth performance and other outcomes related to gut health in both swine and poultry [9–13]. Added protease can also improve digestibility of other nutrients that may be sequestered by proteins, such as starch and fat [12, 14]. The evidence is apparently less clear for the value of exogenous protease when administered in combination with other enzymes or probiotics, as indicated by a recent meta-analysis [15]. Nevertheless, improvements in nutrient digestibility and/or performance when applied to feed in combination with other enzymes/probiotics have been reported for certain proteases [10, 16, 17].

Against this background, the present study used LC-MS and MS/MS to investigate and characterize the *in vitro* substrate specificity (AA cleavage preference) or substrate broadness of a commercially available subtilisin variant from *Bacillus amyloliquefaciens*, referred to herein as FNA protease (EC 3.4.21.62), under varied conditions of pH, digestion time and substrate concentration, using soy protein isolate (SPI) as the model substrate. This protease is efficacious at improving protein and AA digestibility, nutrient utilization and apparent ileal metabolizable energy in corn-soybean meal-based diets with/without added industrial co-products in broilers and pigs [10, 16, 18]. The AA cleavage preferences of related members of this S8 protease family have been studied in the context of a few single proteins and synthetic substrates only, as detailed in the MEROPS peptidase database [19]. They have not previously been studied for complex protein substrates like SPI which contains 90% soy protein besides moisture [20]. Similarly, the cleavage preferences of endogenous porcine pepsin and pancreatin, which have not previously been studied for SPI substrate, were incorporated into the study design. Further, a second commercial chymotrypsin-like feed serine protease derived from *Nocardiopsis prasina*, an S1 family member [19, 21, 22] hereafter denoted as NPP (EC 3.4.21.-), was studied alongside FNA for comparison. Like FNA, NPP has previously been shown to be quite efficacious at improving ileal AA digestibility and growth performance in both pigs and poultry [9, 13, 23]. Using LC-MS for peptide separation and identification with SPI as the substrate, the exogenous and endogenous proteases were tested individually as well as in combination (endogenous plus exogenous protease) to enable exploration of the specific peptide cleavage activity of the commercial (exogenous) proteases, how this differs from that achieved by endogenous proteases alone, and whether there is a complementary effect between the two when applied to the substrate. We performed three experiments in which SPI:protease ratio, reaction pH and time, denaturants, as well as instruments and instrument settings were varied, and used the results to propose two new concepts: 1) NPBCF (Normalized Peptide Bond Cleavage Frequency) to describe a protease's specificity for cleaving peptide bonds formed with different AAs, and; 2) PSBI (Protease Substrate Broadness Index), an enzymatic

parameter describing if it has wider or narrower substrate specificity, *i.e.*, its general capability to cleave all 20 AA formed peptide bonds found in a protein.

## Materials and methods

### Proteases, soy protein isolate, buffers and stock solutions

Purified porcine pepsin (EC 3.4.23.1) containing 2500 units of activity per mg protein was obtained from Roche Applied Science (lot #10108057001). Pancreatin extracted from porcine pancreas was obtained from Sigma-Aldrich (SIGMA #P7545, lot# SLBJ7293V). It had 207 USP (United States Pharmacopeia) protease units, 238 USP α-amylase (EC 3.2.1.1) units and 30 USP lipase (EC 3.1.1.3) units per mg and was able to digest 25 times its weight of casein in 1 h at pH 7.5 40˚C according to the supplier. The protease activity of porcine pancreatin is contributed by both endo- and exopeptidases of trypsin (EC 3.4.21.4), chymotrypsin (EC 3.4.21.1), elastase (EC 3.4.21.71), carboxypeptidase A (EC 3.4.12.2) and B (EC 3.4.12.3) [24]. FNA, a subtilisin variant (27.5 kDa), was obtained from DuPont with a protein concentration of 45 mg/ml. NPP was an unspecific chymotrypsin-like serine endopeptidase (19.1 kDa) from *N. prasina* and expressed in *Bacillus licheniformis* [11, 13, 21, 22]. It was extracted from RONOZYME® ProAct (DSM Nutritional Products, Kaiseraugst, Switzerland) with an adjusted protein concentration of 7 mg/ml. Endoproteinase Glu-C from *Staphylococcus aureus* strain V8 (EC 3.4.21.19) sequencing grade hydrolyzing peptide bonds at the carboxyl side of glutamyl residues was obtained from Calbiochem (Cat#324712, San Diego, CA). Soy protein isolate (SPI) (SUPRO 760 IP; 90% protein content on dry matter basis) and its AA composition (%) were provided by DuPont. Unless otherwise stated all other reagents were of analytical grade and were sourced from Sigma-Aldrich.

SPI stock solution (1 mg/ml) was prepared by dissolving 10 mg SPI in 10 ml of acetate buffer (50 mM, pH 3.7) or ammonium bicarbonate buffer (50 mM, pH 6.5). Pepsin stock solution (200 ng/μl) was prepared in 10 mM HCl. Pancreatin stock solution (100 ng/μl) was prepared by dissolving 1 mg pancreatin in the pH 6.5-buffer. FNA stock solution was prepared by mixing 3 μl FNA concentrate with 1.35 ml pH 6.5-buffer. NPP stock solution was prepared by mixing 10 μl NPP concentrate with 0.7 ml pH 6.5-buffer. All stock solutions were stored at -18˚C until use. Working solutions of each protease were prepared immediately prior to use, by appropriate dilution of stock solutions with the relevant buffer. Working solutions comprised: pepsin 10 ng/μl; pancreatin 5 ng/μl, 10 ng/μl; FNA 5 ng/μl; 10 ng/μl; 100 ng/μl; NPP 5 ng/μl, 10 ng/μl, 100 ng/μl.

### *In vitro* protease digestion of SPI

In Experiment 1 and 2, the digestion of SPI by protease was performed using commercially available microcentrifuge filtration devices (spinfilters) with a molecular mass cut-off of 10 kDa (Vivacon™ 500 DNA Concentrator, p/n VN01H02, Sartorius, Germany). Protein digestions were performed in duplicate for Experiment 1 and in triplicate for Experiment 2, and analyzed by LC-MS in triplicate. Sample preparation was as follows: 20 μl (Experiment 1) or 100 μl (Experiment 2) of SPI stock solution (corresponding to 20 μg or 100 μg protein, respectively) were transferred to a 10 kDa spinfilter inside a spinfilter tube containing 400 μl 8 M urea and left for 20 min at room temperature (22˚C) for protein unfolding. The mixture was then centrifuged at 14000 g for 15 min, repeated after each of two consecutive washes with 100 μl of the buffer pH 3.7 or pH 6.5. The solution-free spinfilter was transferred to a new spinfilter tube and 100 μl of the respective protease (10 ng/μl), or 50 μl (10 ng/μl) plus 50 μl (10 ng/μl) of the respective proteases for the combined digestions (as set out in Table 1), were added to start the reaction. Digestions were incubated at 37˚C for 1 h. The mixture was then

**Table 1. Conditions for the hydrolysis of soy protein isolate (SPI) by endogenous and exogenous proteases at 37˚C for 1 h (Experiment 1–2) and 2 h (Experiment 3).**

| Protease | Substrate-to-protease ratio (w/w) | Reaction volume | pH |
|---|---|---|---|
| *Experiment 1* | | | |
| Pepsin | 20:1 | 100 μl | 3.7 |
| Pancreatin | 20:1 | 100 μl | 6.5 |
| FNA | 20:1 | 100 μl | 3.7 |
| FNA | 20:1 | 100 μl | 6.5 |
| FNA+pepsin | 20:0.5:0.5 | 100 μl | 3.7 |
| FNA+pancreatin | 20:0.5:0.5 | 100 μl | 6.5 |
| *Experiment 2* | | | |
| Pepsin | 100:1 | 100 μl | 3.7 |
| Pancreatin | 100:1 | 100 μl | 6.5 |
| FNA | 100:1 | 100 μl | 3.7 |
| FNA | 100:1 | 100 μl | 6.5 |
| FNA+pepsin | 100:0.5:0.5 | 100 μl | 3.7 |
| FNA+pancreatin | 100:0.5:0.5 | 100 μl | 6.5 |
| *Experiment 3* | | | |
| Pancreatin | 200:1 | 25 μl | 6.5 |
| FNA | 200:1 | 25 μl | 6.5 |
| NPP | 200:1 | 25 μl | 6.5 |
| FNA+pancreatin | 200:1:1 | 25 μl | 6.5 |
| NPP+pancreatin | 200:1:1 | 25 μl | 6.5 |

centrifuged as above, and the run-through fraction collected. The remaining fraction was washed with two 100 μl portions of 0.1% (v/v) trifluoroacetic acid (TFA) and then pooled with the run-through fraction. The pooled fractions were dried at room temperature in a vacuum centrifuge and the vacuum dried material was collected and re-dissolved in 50 μl 0.1% (v/v) TFA. A 2 μl aliquot was injected into the nano LC-MS (Orbitrap Fusion™ Tribrid™ from Thermo Scientific, Waltham, MA, USA) for peptide separation, identification and quantification. Blanks comprised of solvent only without SPI or protease.

In Experiment 3, digestion of SPI by protease was performed using the S-Trap™ digestion procedure (ProtiFi, NY, USA) and S-trap™ micro columns (p/n $CO_2$-micro-80; ProtiFi, NY, USA). Protein digestions were performed and analyzed in triplicate. Sample preparation was as follows: 25 μl of SPI stock solution having 25 μg protein was transferred to an Eppendorf Protein LoBind tube containing 25 μl of SDS-TEAB (10% w/v SDS, 100 mM triethylammonium bicarbonate pH 7.5) and heated at 95˚C for 10 min for protein unfolding. The LoBind tube was then cooled to room temperature and 5 μl of 12% (v/v) phosphoric acid were added to give a final concentration of 1.2% phosphoric acid. S-Trap™ binding buffer 330 μl consisting of 100 mM TEAB (pH 7.1) and 90% methanol was then added to the LoBind tube and the resulting solution was added sequentially in three equal portions of 128.5 μl to the S-Trap, followed by centrifuging each time at 4000 g for 2 min until all the solution had passed through. The S-Trap was then washed three times with 150 μl of the S-Trap™ binding buffer by centrifugation. The solution-free S-Trap™ was then placed in a new tube into which 25 μl of respective protease (5 ng/μl), or 12.5 μl (10 ng/μl) plus 12.5 μl (10 ng/μl) of the respective proteases for the combined digestions, was added to start the digestion. Digestions were carried out at 37˚C for 2 h. Generated peptides in the digestion were eluted by sequential addition of 40 μl 50 mM TEAB (pH 8.9), 40 μl 0.2% formic acid (FA), and 35 μl acetonitrile/$H_2O$/FA (50:50:0.2), with a

centrifugation step (4000 g for 2 min) between each elution. The obtained peptides were dried down and resuspended in 50 μl 0.1% TFA. A 2 μl aliquot was injected into the nano LC-MS system (Q Exactive ™ HF Quadrupole-Orbitrap™ from Thermo Scientific). An additional experiment was carried as Experiment 3 testing the digestion of SPI by Endoproteinase Glu-C, *i.e.*, SPI:Glu-C (200:1) and SPI:pancreatin:Glu-C (200:1:1) at pH6.5 for 2 h.

### An overview of the experimental design and associated *in vitro* protein hydrolysis conditions

Three sequential experiments were carried out to investigate the peptide cleavage preference and substrate broadness of selected endogenous (pepsin, pancreatin) and exogenous (FNA, NPP) proteases using SPI as the protein substrate (Table 1).

### LC-MS separation and analysis of SPI hydrolysis products

The LC-MS instrument settings and separation conditions for each experiment are given in Table 2. Control (blank) samples comprising buffer (pH 3.7 and pH 6.5) were run alongside the protease treated samples. All determinations were in triplicate.

### LC-MS data processing

Spectra of individual peptides generated by the LC-MS were identified by comparison with known soy proteins using the Mascot search engine (version 2.6.2.0; Matrix Science, London,

**Table 2. LC-MS conditions for separating and analyzing the hydrolysis products of SPI by endogenous and exogenous proteases.**

| Analytical system | Instrument | Parameter | Conditions |
|---|---|---|---|
| LC | Ultimate 3000 Nano LC system (Thermo Scientific) | Pre-column | Acclaim PepMap 100 C18 100 Å, 5 μm (100 μm i.d. x 200 mm) |
| | | Column | EASY-column C18-A2, 3 μm (75 μm i.d. x 1000 mm) |
| | | Mobile phase | A: $H_2O$ 99.9%: Formic acid 1% |
| | | | B: Acetonitrile 99.9%: Formic acid 1% |
| | | Flow rate | 0.300 μL/min |
| | | Column temperature | 50˚C |
| | | Run time | 65 min |
| | | Injection volume | 2 μL |
| MS (Experiment 1 and 2) | Orbitrap Fusion™ Tribrid™ | Standard | |
| | | Ion source | NSI |
| | | Polarity | Positive |
| | | Collision gas | $N_2$ |
| | | Collision type | HCD |
| | | Ions pray voltage | 2500 V |
| | | Ion transfer tube | 275˚C |
| | | Scan event 1 | FTMS + p NSI Full MS [350–1550] |
| | | Scan event 2–3 | FTMS + p NSI d Full MS/MS (DDA) |
| MS (Experiment 3) | Q Exactive™ HF Quadrupole-Orbitrap™ | Standard | |
| | | Ion source | NSI |
| | | Polarity | Positive |
| | | Collision gas | $N_2$ |
| | | Collision type | HCD |
| | | Ion spray voltage | 2500 V |
| | | Ion transfer tube | 250˚C |
| | | Scan event 1 | FTMS + p NSI Full MS [350–2000] |
| | | Scan event 2–3 | FTMS + p NSI d Full MS/MS (DDA) |

UK). Information and parameters on the identified peptides including UniProt accession number, monoisotopic m/z, calculated/theoretical mass, start and end residue number in candidate soy protein sequence, AA residue before the identified sequence, the identified peptide sequence (of which the last residue is P1), and AA residue after the identified sequence (P1') are deposited as S1–S3 Tables under Support information. These identified peptides were sorted by their C-terminal AA residue at P1 position according to the nomenclature by Schechter and Berger, who named the AA residues left of the scissile bond as P1-P4 while right of the scissile bond as P1'-P3' in the study of papain [26], and counted with duplicates removed.

The size distribution of identified peptides in the SPI digestion by the proteases was illustrated as the number of peptides within a range of m/z of 0.1 kDa, relative to the total number of identified peptides. Data are presented as means of two (Experiment 1) or three (Experiments 2 and 3) biological replicates. All LC-MS determinations were performed in triplicate, and resulting values were averaged.

## Results

### Protease hydrolysis of SPI at pH 3.7/1 h and the determination of NPBCF and PSBI

The AA composition (%) of the SPI used in this study is given in Column A of Table 3 with all 20 AAs represented. The AA composition is also indicative of the abundancy of the peptide bonds for each AA in the SPI. The amounts of Glu, Gln, Asp and Asn were estimated from the amounts of Glu+Gln and Asp+Asn provided by the vendor and the Glu/Gln, Asp/Asn ratios from genome sequencing data of soy product [25]. Table 3 further shows the calculation of the parameters of NPBCF and PSBI with the hydrolysis of SPI at pH3.7 by FNA as an example. Thus, in Column B the first number of 94 means average of 94 peptides with different lengths but all with Ala (P1) at their C-termini were detected under the experiment setup of Experiment 1. The generation of the 94 peptides all with Ala at P1 also indicates FNA had less selectivity on the AAs residues in SPI flanking the P1 of Ala, *i.e.*, P4-P2, P1-P4, whose identities can be found in S1 Table. To compare and examine the cleavage preferences of the selected proteases and their combinations, NPBCF of the 20 AA-formed peptide bonds was defined and calculated as shown Column D to F in Table 3. Peptide Bond Cleavage Frequency (PBCF) in Column D was determined as the number of peptides generated having a common AA residue at P1 divided by the total number of peptides generated in the digestion and multiplied by 100. PBCF with respect to a given AA formed peptide bond was divided or normalized by the percentage of this AA in SPI (*i.e.*, the relative concentration of this AA bonds in SPI) and expressed as percentage (Column E). NPBCF (Column F) is the percentage of Column E divided by the sum of Column E. Protease Substrate Broadness Index (PSBI) is defined as the reciprocal of the standard deviation (Stdev) of the mean NPBCF of all AAs found in SPI multiplied by 100.

Fig 1 shows the total number of peptides generated from SPI following incubation with FNA, pepsin or FNA+pepsin, during a 1 h digestion at pH 3.7 (37˚C) and SPI-to-protease ratio of 20:1 or 100:1. When administered alone, the subtilisin typed FNA having an optimal pH of around 8.4 generated 59% and 39% of the number of peptides produced by the acidic protease pepsin with a pH optimum of 1.0–4.0, at a SPI-to-protease ratio of 20:1 and 100:1, respectively (Fig 1). Increasing substrate availability from SPI-to-protease ratio of 20:1 to 100:1, the total number of peptides generated decreased by about 49 to 67% for FNA, pepsin, and their combination (Fig 1).

**Table 3. SPI AA composition, the calculation of NPBCF, mean NPBCF, its standard deviation (Stdev), and the reciprocal of the Stdev timed 100 (PSBI).** Reaction conditions: SPI:FNA ratio of 20:1 at pH 3.7, 37°C 1 h (Experiment 1).

| Column | A | B | C | D | E | F |
|---|---|---|---|---|---|---|
| Amino Acid (AA) at P1† | SPI AA composition (%) | No. of peptides with the same AA at C-terminal (n = 2) | Stdev of the No. of peptides | PBCF (= B/ 1288.5*100) | Normalized against SPI AA composition (= D/A) | NPBCF (= E/ 17.8*100) |
| Alanine, Ala | 4.3 | 94 | 5.8 | 7.3 | 1.7 | 9.5 |
| Cysteine, Cys | 1.3 | 3 | 1.2 | 0.2 | 0.2 | 1.0 |
| Aspartic acid, Asp | 5.7 | 44.5 | 0.6 | 3.5 | 0.6 | 3.4 |
| Glutamic acid, Glu | 10.0 | 190 | 1.2 | 14.7 | 1.5 | 8.3 |
| Phenylalanine, Phe | 5.2 | 85 | 0.0 | 6.6 | 1.3 | 7.1 |
| Glycine, Gly | 4.2 | 20.5 | 0.6 | 1.6 | 0.4 | 2.1 |
| Histidine, His | 2.6 | 8 | 0.0 | 0.6 | 0.2 | 1.3 |
| Isoleucine, Ile | 4.9 | 16.5 | 1.7 | 1.3 | 0.3 | 1.5 |
| Lysine, Lys | 6.3 | 127 | 3.5 | 9.9 | 1.6 | 8.8 |
| Leucine, Leu | 8.2 | 166.5 | 4.0 | 12.9 | 1.6 | 8.9 |
| Methionine, Met | 1.3 | 25 | 1.2 | 1.9 | 1.5 | 8.4 |
| Asparagine, Asn | 6.0 | 91 | 4.6 | 7.1 | 1.2 | 6.6 |
| Proline, Pro | 5.1 | 17 | 1.2 | 1.3 | 0.3 | 1.5 |
| Glutamine, Gln | 9.1 | 143 | 5.8 | 11.1 | 1.2 | 6.8 |
| Arginine, Arg | 7.6 | 62 | 2.3 | 4.8 | 0.6 | 3.6 |
| Serine, Ser | 5.2 | 52.5 | 0.6 | 4.1 | 0.8 | 4.4 |
| Threonine, Thr | 3.8 | 58 | 2.3 | 4.5 | 1.2 | 6.7 |
| Valine, Val | 5.1 | 23 | 1.2 | 1.8 | 0.4 | 2.0 |
| Tryptophan, Trp | 1.3 | 5 | 0.0 | 0.4 | 0.3 | 1.7 |
| Tyrosine, Tyr | 3.8 | 57 | 3.5 | 4.4 | 1.2 | 6.5 |
| Sum | 101.0 | 1288.5 | 5.2 | 100.0 | 17.8 | 100.0 |
| Average | 5.0 | n/a | n/a | 5.0 | n/a | 5.0 |
| Stdev | n/a | n/a | n/a | n/a | n/a | 3.0 |
| PSBI | n/a | n/a | n/a | n/a | n/a | 33.3 |

†P1, first AA residue left of the scissile peptide bond [26]. n/a., not applicable.

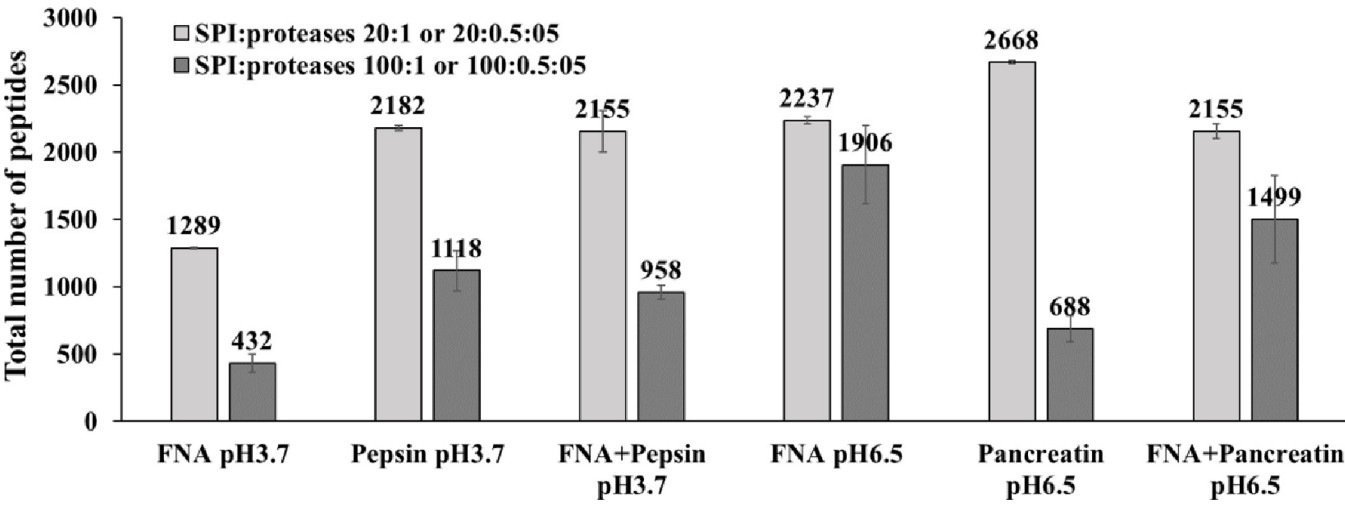

**Fig 1. Number of peptides generated from SPI by FNA, porcine pepsin, pancreatin and their combinations.** Conditions: pH 3.7 and 6.5, 37°C 1 h, soy protein isolate (SPI):protease ratio of 20:1 or 20:0.5:05, n = 2, and 100:1 or 100:05:0.5, n = 3.

**A)**

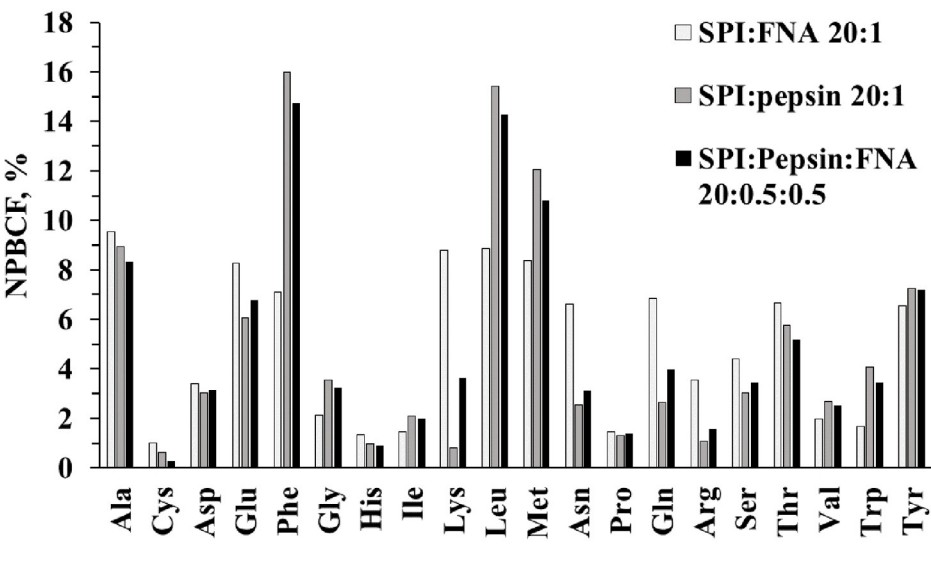

**B)**

**Fig 2. NPBCF of SPI by FNA, porcine pepsin and their combinations.** Conditions: pH 3.7, SPI:protease ratio of 20:1 (A), n = 2, and 100:1 (B), n = 3.

The NPBCF of each protease or protease combination, calculated as exemplified in Table 3 for FNA, is presented in Fig 2. At pH 3.7, across both SPI-to-protease ratios, pepsin was generally capable of hydrolyzing peptide bonds containing any of the 20 AAs at the P1 residue

position, though the NPBCF values differ greatly among the 20 AAs (Fig 2A and 2B). It is apparent that pepsin was comparatively poor at hydrolyzing bonds containing alkaline amino acids (Lys, Arg and His), Pro, and Cys (under non-reducing conditions) at position P1 (NPBCF < 3%; Fig 2A and 2B). Like pepsin FNA was shown to have a wide spectrum activity capable of cleaving all the 20 AAs formed peptide bonds (Fig 2). There are, however, clear peptide bond cleavage preference differences between pepsin and FNA (Fig 2A and 2B). Thus, FNA was about 2 times higher in NPBCF than pepsin for Lys and Gln. The combination of pepsin and FNA produced an AA cleavage profile broadly similar to that of pepsin alone, except an improved cleavage or NPBCF for Lys and Gln at P1 at both SPI:protease ratios, thus making the NPBCF for all the 20 AA more even (*i.e.*, lower Stdev of mean NPBCF) than pepsin alone. If a protease would have an equal NPBCF for all 20 AA formed peptide bonds then the NPBCF for each AA would be 5.0%, *i.e.*, equal to the mean NPBCF (Table 3). Even though FNA produced some 40% to 60% of the total number of peptides produced by pepsin at either SPI:protease ratios (Fig 1), the molecular size distribution of these peptides generated by pepsin and FNA alone, or in combination was quite close. The percentages of small peptides with a mass of ≤1.2 kDa (equivalent to approximately 10 AA peptides) generated by FNA (38%, 47%) and FNA+pepsin (35%, 43%) had a trend to be slightly higher than pepsin alone (34%, 39%) at SPI:protease ratios of 20:1 and 100:1, especially at 100:1.

## Protease hydrolysis of SPI at pH 6.5/1 h

At pH 6.5, FNA generated 0.84 and 2.81 fold of the number of peptides produced by pancreatin when SPI:protease was 20:1 and 100:1 (Fig 1), respectively. FNA+pancreatin each at half dose generated 2.2-fold more peptides than pancreatin alone at a SPI-to-protease ratio of 100:1 (Fig 1). At pH 6.5 the total number of peptides generated by FNA was about 1.7 (20:1) and to 4.4 (100:1) times greater than those produced at pH 3.7 (Fig 1). Increasing substrate availability from 20:1 to 100:1 (SPI:protease) led to a decrease in the number of peptides produced by FNA, pancreatin and FNA+pancreatin, which was in line with what was observed at pH 3.7 with FNA and pepsin (Fig 1). It is noteworthy that the decrease by FNA was the lowest (15%), by pancreatin the highest (74%), while by a combination of FNA with pancreatin was somewhere in-between (30%) (Fig 1).

While Fig 1 shows the number of peptides released by these proteases and their combinations at pH 6.5, Fig 3A and 3B present the size distributions of these peptides. At a SPI:protease of 20:1, the peptides generated by pancreatin were of higher average mass (size distribution skewed to the right) than those generated by FNA alone or by pancreatin+FNA (size distributions skewed to the left) (Fig 3A). The percentages of small peptides (mass of ≤1.2 kDa) were 37% for pancreatin and 59% for both FNA and FNA+pancreatin (Fig 3A). Similar numbers (pancreatin 34%, FNA 56%, FNA+pancreatin 63%) were also seen at a SPI:protease of 100:1 (Fig 3B).

At pH 6.5, across both SPI:protease ratios of 20:1 and 100:1, both FNA and pancreatin were generally efficient at hydrolyzing peptide bonds containing a broad range of AAs at P1 (Fig 4A and 4B). FNA tended to be more efficient than pancreatin at cleaving Phe, His, Leu, Met and Trp at both SPI: protease ratios in terms of the NPBCF values. For these AA residues, the combination of FNA and pancreatin each at half dose was in general also superior to pancreatin alone (Fig 4A and 4B).

## Protease hydrolysis of SPI at pH 6.5/2 h

Under conditions of pH 6.5, SPI:protease 200:1 for the single enzyme treatment and 200:1:1 for the combination treatment, and a longer incubation period (2 h), the number of peptides

A)

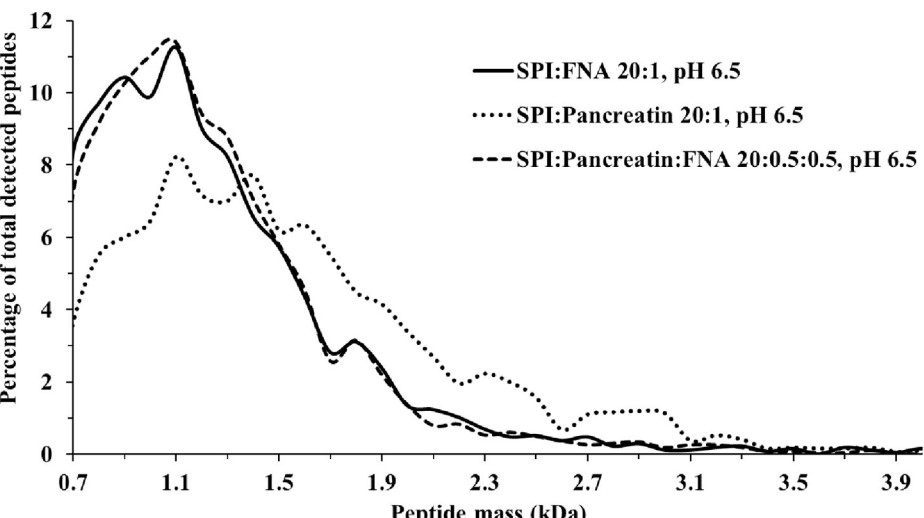

B)

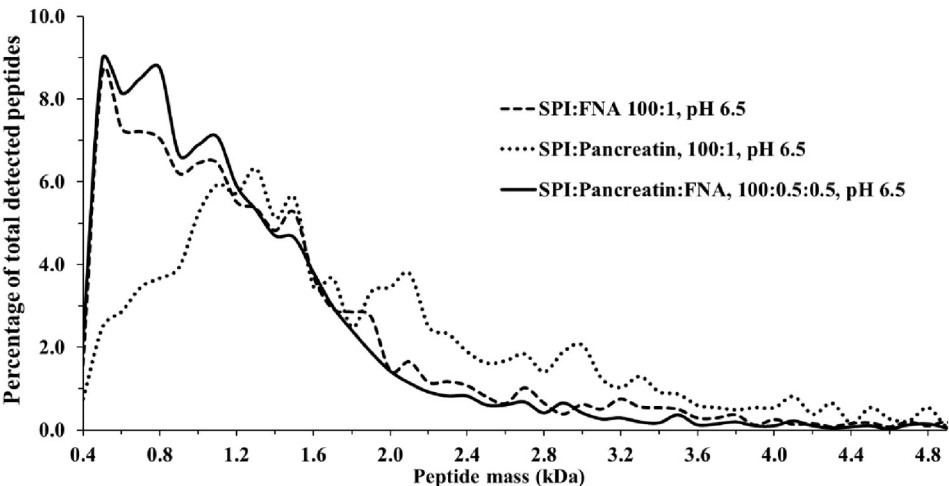

**Fig 3. Size distribution of peptides cleaved from SPI by FNA, pancreatin and their combinations.** Conditions: pH 6.5, 37°C for 1 h, SPI:protease ratio of 20:1 (A), n = 2, and 100:1 (B), n = 3.

generated by pancreatin, NPP, FNA and their combination was, in a decreasing order: pancreatin+NPP (1349); pancreatin+FNA (1096); NPP (1102); FNA (646), pancreatin (450) (Fig 5). The numbers of peptides generated by the protease combinations were similar to the sums of those generated by the proteases administered alone.

**A)**

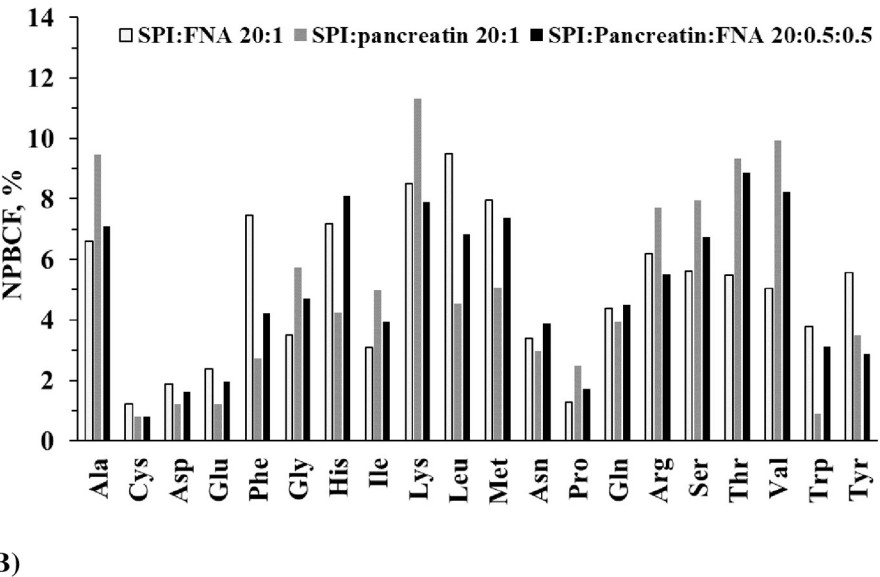

**B)**

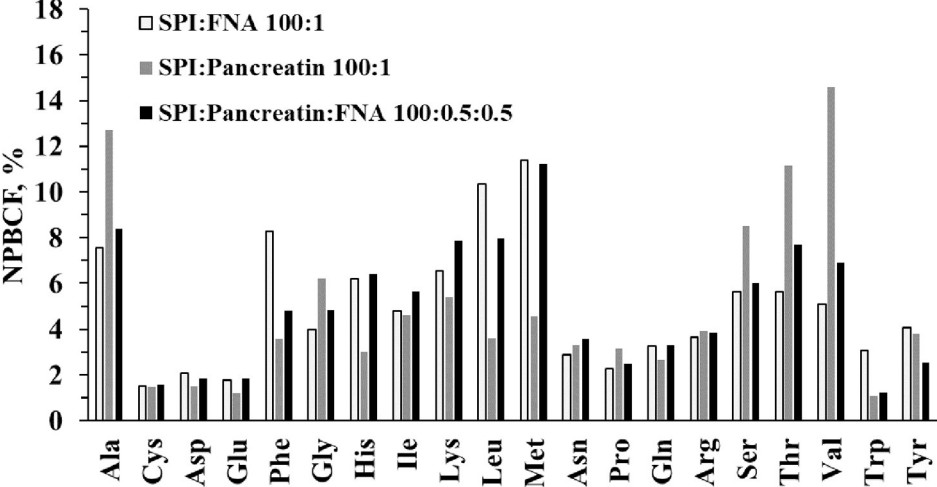

**Fig 4. NPBCF of SPI by FNA, porcine pepsin and their combinations.** Conditions: pH 6.5, SPI:protease ratio of 20:1 (A), n = 2, and 100:1(B), n = 3.

The size distributions of the peptides shown in Fig 5 following incubation of SPI with pancreatin, FNA, NPP and their combinations at pH 6.5, 37°C for 2 h are presented in Fig 6. The size distributions by FNA or FNA+pancreatin skewed to the left than that produced by pancreatin, NPP, or NPP+pancreatin. The percentages of small peptides (mass ≤ 1.2 kDa) generated from these reactions were 20% (pancreatin), 28% (NPP), 44% (FNA), 26% (pancreatin+NPP), and 37% (pancreatin+FNA).

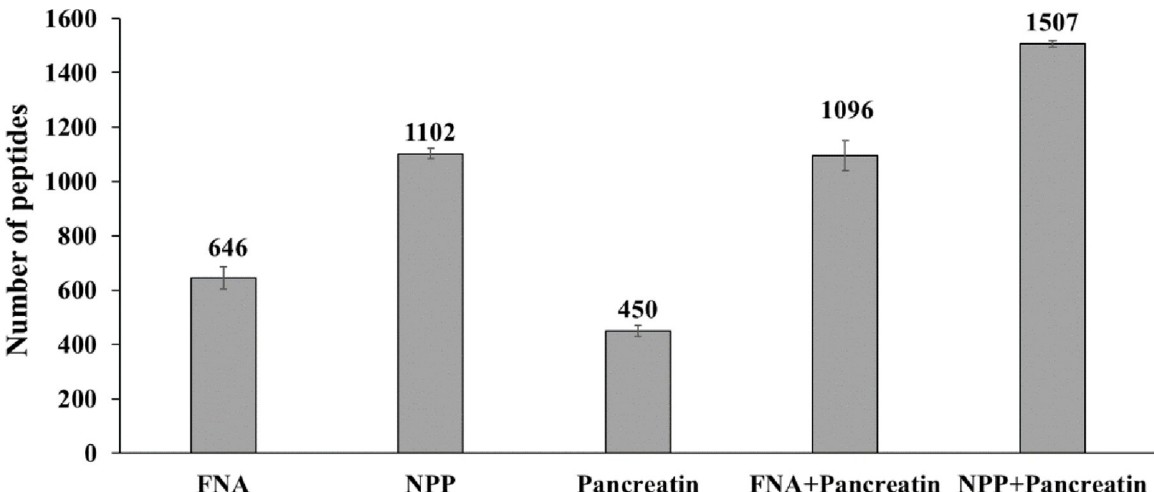

**Fig 5. Number of peptides generated from SPI by FNA, NPP, pancreatin and their combinations.** Conditions: pH 6.5, 37˚C, for 2 h, SPI:protease ratio 200:1 (FNA, NPP, pancreatin) or 200:1:1 (pancreatin+FNA, pancreatin+NPP), n = 3.

Under these experimental conditions, all three proteases (FNA, pancreatin, NPP) and their combinations were effective at hydrolyzing bonds containing a broad range of AAs, but the NPBCF varied markedly across individual AAs (Fig 7). Most strikingly was the high preference

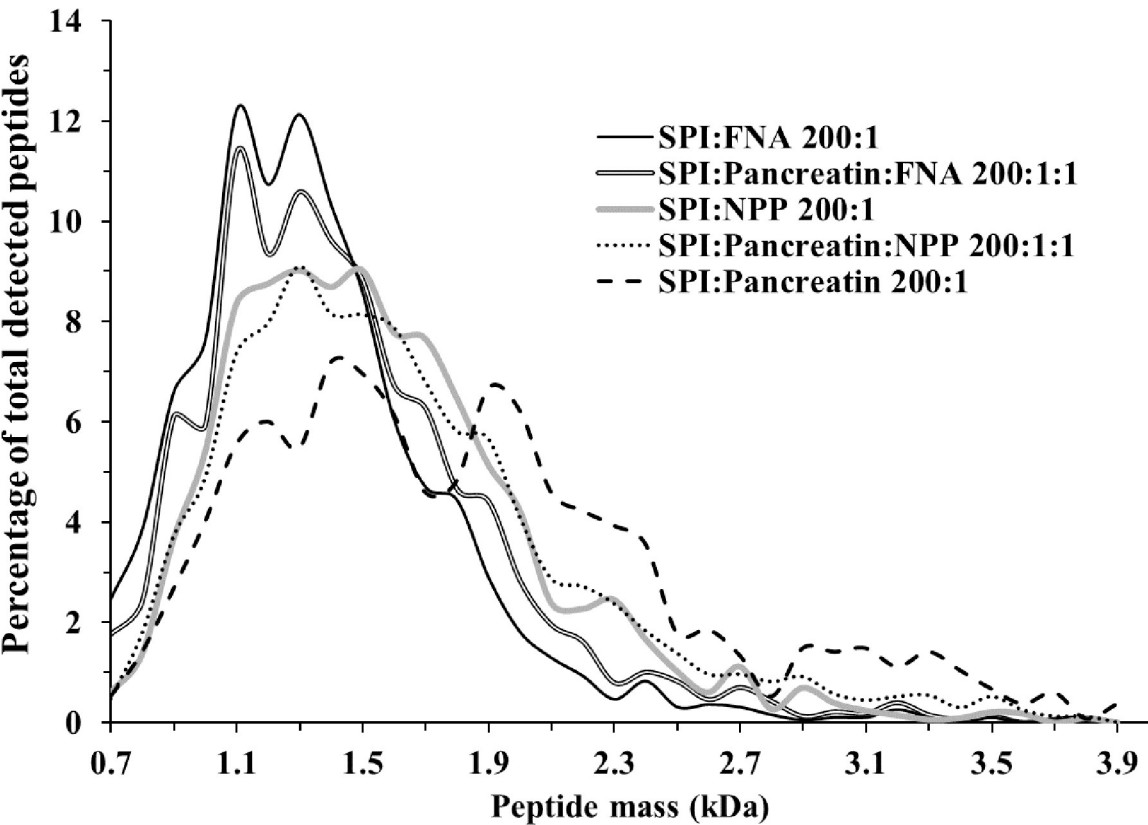

**Fig 6. Size distribution of peptides cleaved from SPI by FNA, NPP, pancreatin and their combinations.** Conditions: pH 6.5, SPI: protease ratio 200:1 (FNA, NPP, pancreatin) or 200:1:1 (pancreatin+FNA and pancreatin+NPP), n = 3.

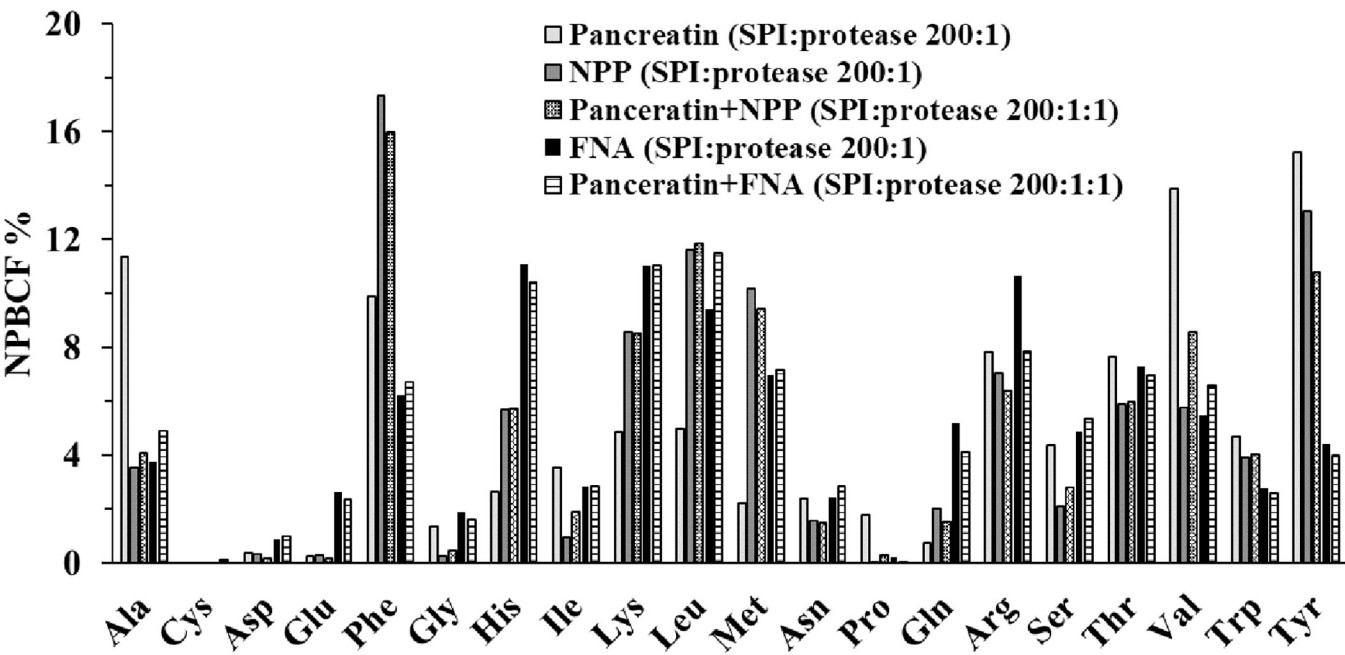

**Fig 7. NPBCF of SPI by FNA, NPP, pancreatin and their combinations.** Conditions: pH 6.5, 37°C, for 2 h, soy protein isolate (SPI):protease ratio 200:1 (FNA, NPP, pancreatin) or 200:1:1 (pancreatin+FNA and pancreatin+NPP), n = 3.

for Phe and Tyr at P1 by both pancreatin and NPP and their combination as indicated by the high NPBCF. Beside Phe and Tyr, pancreatin preferred also Ala and Val at P1. As SPI-to-protease ratio of 20:1 and 100:1, FNA had preference at P1 for His, Leu, Met compared to pancreatin. In addition, FNA also showed higher NPBCF for Glu and Gln compared to both pancreatin and NPP (Fig 7). From Fig 7 it can hence be seen that in general the cleavage preferences of FNA and FNA+pancreatin for each of the 20 individual AAs appeared to be more evenly distributed (*i.e.*, lower Stdev of mean NPBCF) compared with the cleavage preferences of pancreatin, NPP and NPP+pancreatin. It should, however, be noted from Fig 7 that Cys or Pro at P1 in SPI was preferred by neither of the proteases or combinations except pancreatin alone that showed higher cleavage for Pro under the *in vitro* digestion conditions.

In summary of Experiment 1, 2 and 3, though having different preferences on individual AA as P1 as expressed in different NPBCF values, all the exo- and endo- proteases examined had binding pockets capable of accommodating any of the 20 AAs found in SPI as P1 (Figs 2, 4 and 7). Since these 20 AA residues as P1 are found in the protein sequence of SPI, the generation of these peptides shown in Figs 1 and 5 was hence the results that these proteases were able to accommodate any AAs (P4-P2, P1-P4) flanking the P1 AA. The AAs at P4-P2 and P1-P4 for a given P1 can be found in the SPI sequences given in S1–S3 Tables where P1 is specifically listed, which can be any of the 20 AAs for all these proteases examined.

## Mean NPBCF of 20 AAs, its Stdev, and the concept of Protease Substrate Broadness Index (PSBI)

Fig 8A shows the Stdev of the calculated NPBCF values for the 20 AAs, for each protease and protease combination, under each of the tested conditions of pH, SPI:protease ratio, and digestion duration (1 or 2 h). An example of calculating the Stdev of NPBCF is given in Table 3. The magnitude of the calculated Stdev for a given protease varied with the digestion conditions. For

A)

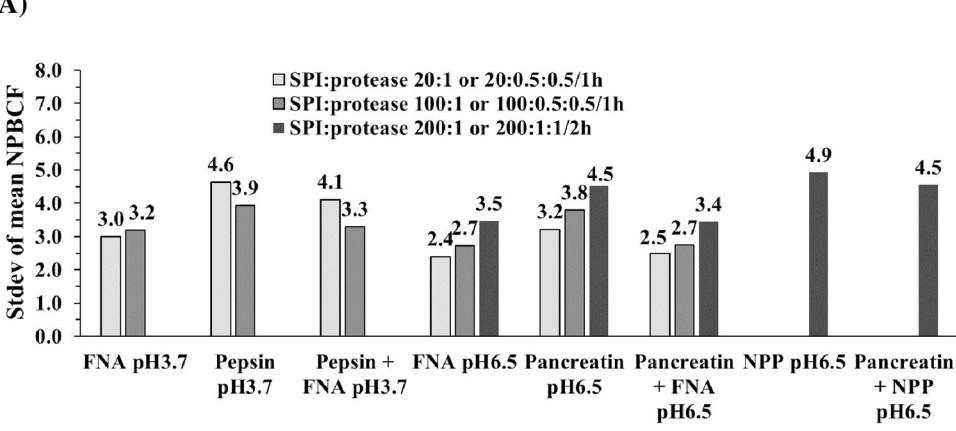

B)

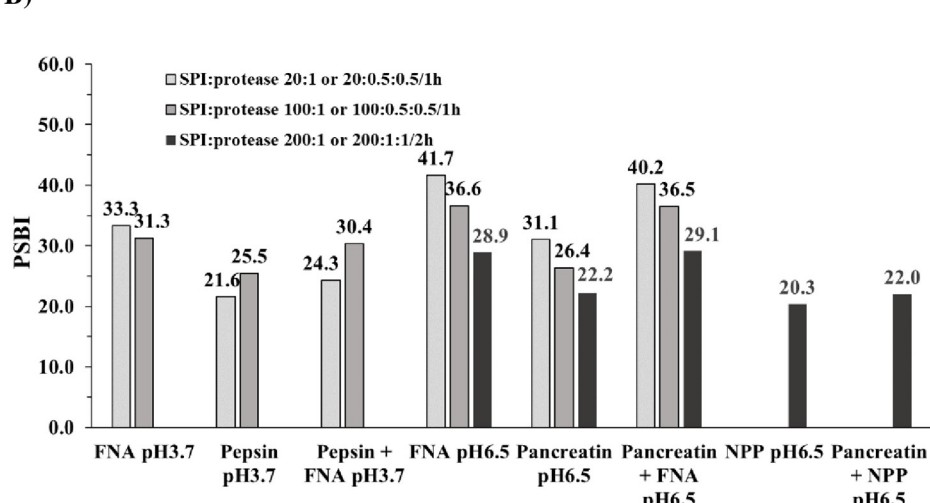

**Fig 8. Standard deviation of mean NPBCF and PSBI achieved by different proteases and their combinations.**
Conditions: pH 3.7 and pH 6.5 at 37°C, for 1 or 2 h at varied SPI:protease ratio.

example, the NPBCF Stdev produced by FNA at pH 6.5 were 2.4, 2.7, and 3.5 with an SPI:protease ratio of 20:1 (1 h), 100:1 (1 h) and 200:1 (2 h), respectively. Despite this variation in response across digestion conditions, FNA produced the lowest NPBCF Stdev relative to those produced by the other proteases and protease combinations under all tested conditions (Fig 8A). For instance, at pH 3.7, the NPBCF Stdev produced by FNA was 1.6 and 0.7 percentage point (p.p.) lower than that produced by pepsin, when the SPI:protease ratio was 20:1 and 100:1, respectively, and the combination of FNA+pepsin produced a NPBCF Stdev that was 0.6 p.p. lower than that produced by pepsin alone. A similar pattern was seen at pH 6.5 for FNA compared with pancreatin or NPP. When FNA was administered in combination with pancreatin, the NPBCF Stdev was reduced by 1.1 p.p. compared to pancreatin alone, whereas when NPP was administered in combination with pancreatin, the Stdev remained the same as pancreatin alone.

PSBI is defined as the reciprocal of the Stdev of mean NPBCF multiplied by 100 (Table 3). The PSBI for each protease and protease combination is shown in Fig 8B. Under all the tested

incubation conditions (pH, SPI:protease ratio and incubation time, different sample preparation and LC-MS instruments), the PSBI of FNA was higher than those of the other proteases. When FNA was administered in combination with endogenous pepsin or pancreatin, the PSBI was higher than when these endogenous proteases were administered alone: the PSBI of FNA +pepsin was 2.7 p.p. (20:1) and 4.9 p.p. (100:1) higher than that of pepsin alone; the PSBI of FNA+pancreatin was 9.1 p.p. (20:1), 10.1 (100:1) and 6.9 p.p. (200:1) higher than pancreatin alone. In contrast, NPP did not have the effect of FNA since the PSBI produced by NPP+-pancreatin (22.0) was not higher than pancreatin alone (22.2) (Fig 8B). Under the conditions of Experiment 3, Endoproteinase Glu-C, which is more specific than FNA and NPP, was examined for both NPBCF and PSBI. At pH6.5 by NPBCF, Glu-C was 15 times more active on Glu than Asp in SPI at P1. Combination of pancreatin with Glu-C (SPI:Glu-C:pancreatin of 200:1:1) improved pancreatin's capability to hydrolyze Glu at P1 in SPI by a factor of 10. Glu-C had a PSBI of 11.0 on SPI, which is two times lower than that what was found for pancreatin (Fig 8B).

## Discussion

Maximizing animal feed protease efficacy in improving protein digestibility *in vivo* relies on a good understanding of the extent of complementation between the activity of the exogenous feed protease and that of the endogenous proteases in the digestion of major feed protein ingredients. *In vitro*, predictions about the likely efficacy of feed proteases have tended to rely heavily on protease activity measurements in terms of initial velocity on peptide bond hydrolysis where the percentage of protein substrate hydrolyzed during the digestion duration is usually less than 5%. This means only those most preferred peptide bonds in a protein substrate are hydrolyzed in the activity assay. This is because a protein has many types of peptide bonds and each peptide bond can be regarded as a potential substrate, and the peptidic products generated by protease hydrolysis can function again as new substrates. Thus, measurement of hydrolysis activity does not reveal information of the substrate broadness of a protease, *i.e.* the 'capability' of the protease for cleaving any peptide bonds within the substrate and has difficulty in assessing possible complementation effects between proteases. These challenges were tackled in the current study by LC-MS counting the number of unique peptides generated (*i.e.*, with duplicated peptides removed). Individual proteases can vary markedly in their specificity for AA sequences at and neighboring the P1 residue; whilst some will cleave a variety of sequences, others are highly sequence specific (https://www.ebi.ac.uk/merops/). It is therefore hypothesized that, in addition to hydrolysis activity measured in terms of initial reaction velocity, the 'cleavage preference' of a protease may be a significant indicator of its likely *in vivo* efficacy in a given dietary setting and such "cleavage preference" should be also quantifiable, which led to the proposal of NPBCF and PSBI derived from the Stdev of mean NPBCF. Both NPBCF and PSBI are enzymatically biochemical parameters like enzymes' optimal pH and temperature values that may vary with assay conditions (Fig 8A and 8B). In the current study three experiments with varied reaction conditions, LC-MS models and their setups were carried out to generate data based on which these two new notions were proposed. These include reaction pH 3.7 and 6.5 to mimic the pH values of the stomach filled with digesta and small intestine of monogastric animals [27, 28], substrate to protease ratios, reaction time and the use of different denaturants to facilitate protein hydrolysis. Before the start of protease digestion, SPI was unfolded by one of the two denaturants: urea breaking hydrogen bonding (Experiment 1–2), SDS disrupting hydration shell and expanding the hydrophobic core of proteins (Experiment 3). Before the addition of protease to start the digestion, the denaturants were removed by 10 kDa membrane filtration to avoid their possible effect on the proteases. It

could, however, happen that certain SPI components might have undergone renaturation in the *in vitro* protease digestion steps. It should be noted *in vitro* experiments including the one described here, though possibly a good way to reveal mechanisms of protein digestion *in vivo*, it may certainly not replace animal trials because it does not provide parameters of animal performance such as average daily body weight gain and feed conversion ratio.

At pH 3.7 FNA had a PSBI value of 33.3 whilst at pH 6.5 it had a PSBI value of 41.7 (Fig 8B). PSBI is not only a parameter that describes a protease's substrate broadness, but it is also useful in assessing the effect of two or more proteases used together on their joint substrate broadness. Thus, FNA when used together with pancreatin improved pancreatin's PSBI from 22.2 to 29.1, whereas under the same conditions, chymotrypsin-like NPP gave no improvement (Fig 8B). This may be explained by the observation that NPP and pancreatin had overlapping peptide preferences (Fig 7) and by the fact that NPP is a member of S1 protease family where the endogenous proteases trypsin, chymotrypsin and elastase (EC 3.4.21.36) in the pancreatin belong, whilst the subtilisin-typed FNA is a S8 protease family member [19].

To determine PSBI, NPBCF must be determined first since PSBI is the reciprocal of Stdev of mean NPBCF; the sequence of the protein substrate needs to be known, duplicated peptides need to be removed from the total peptide pools LC-MS detected and grouped according to their C-terminal AA (P1 position) as demonstrated in Table 3. For example, FNA at pH 3.7, 37˚C 1 h in a SPI:FNA ratio of 20:1 generated 94 peptides with different masses but all had Ala at their C-termini (Table 3). In defining PSBI even though the data on the AAs at position P1 of the scissile bond in SPI are weighed, it should be mentioned that the AAs positioned at P4-P2 and P1'-P4' which can be found in S1–S3 Tables are not neglected. This is because complex substrate of SPI had been used and the number of peptides generated (Figs 1 and 5) by each of the proteases and protease combinations from SPI was unavoidably affected by AAs on either side of the scissile bond and even by the presence of secondary structures like disulfide bonds these AAs could be involved in. The concept of the PSBI could be primarily useful for the assessment of animal feed proteases, detergent proteases, and proteases used in the hydrolysis industries, *i.e.*, industrial proteases with a lower bias toward AAs near the scissile bonds being desirable. It can be envisaged that more specific proteases will have lower PSBI values. Under the conditions of Experiment 3 Endoproteinase Glu-C from *S. aureus* strain V8 which favors Glu at P1 generated four times less peptides than pancreatin and had a PSBI value of 11.0, two times lower than pancreatin. It may also be stressed that substrates to be used for the estimation of PSBI should be proteins having known AA sequences preferably with all 20 AAs well represented. Synthetic substrates like succinyl-AAPX-pNA, which have been used in determining the most favored AA at P1 for NPP homologue [22], are not suitable.

The total number of peptides generated varied from protease to protease and varied also with SPI:protease ratios (Figs 1 and 5), usually lower number of peptides being produced at higher SPI:protease ratios. This is because at higher SPI concentration, proteases tend to hydrolyze their most preferred peptide bonds first, generating more duplicated peptides that were removed in the data processing step. In the case of pancreatin increased SPI ratios also means increased ratio of soybean trypsin and chymotrypsin inhibitors to protease [19]. As reported earlier [29], the total number of peptides generated under the same reaction conditions also varied in the replicate digestions (Figs 1 and 5). In the current work duplicates (20:1) gave good reproducibility but at higher ratio of 100:1 and 200:1 triplicates were chosen (Figs 1 and 5).

Besides the difference in total number of peptides generated for each protease and their combinations, another feature of the pool of peptides is their size distribution. At pH 3.7 the size distribution of the pool of peptides generated by FNA resembled pepsin and pepsin+FNA with peptide mass ≤1.2 kDa in the range of 34–47%, but at pH 6.5 FNA or FNA+pancreatin

constantly generated 16–22% more portions of small sized peptides than pancreatin, NPP or pancreatin+NPP (Figs 3 and 6). Small sized peptides with masses of less than 1.2 kDa containing approximately 10 AA may be easier to diffuse into the mucosa layer in the small intestine and are also the suitable sizes for membrane anchored peptidases to process to AA, di- and tri-peptides, which are transported inside the epithelial cells [30, 31]. Peptides of smaller size (average chain length 3.2) have been shown to be absorbed more rapidly than larger peptides [32].

The substrate specificity of a protease is usually studied by using a range of synthetic substrates with the change of AA at P1 to P4 or in some cases also P1'-P4'. However, synthetic substrates do not represent well a protease's natural substrate with respect to secondary structures and posttranslational modifications, such as disulfide formation, N-glycosylation of Asn, phosphorylation of Ser and Tyr which all occur in SPI used in the current study. Traditionally, oxidized insulin B chain has been used to assess protease peptide cleavage preference and subsequently, myoglobin has been used in the evaluation of peptide bond cleavage preference of pepsin and fungal aspartic proteases [33]. At start of this study, we also tested myoglobin and BSA as substrates. It proved difficult to evaluate the NPBCF of these AAs at P1, *i.e.*, Asn, Tyr, Trp, Arg and Cys in myoglobin and Met and Trp in BSA, due to their low contents (1% or less) (our unpublished data). This led to the choice of SPI, which is a mixture of proteins found in soy bean seed having a balanced AA composition (Table 3), widely used in food and feed, and additionally mass data for soybean proteins are now available.

In nature there are no proteins having the same AA content (5.0%) for each of the 20 AAs and this is also the case for SPI having highest amount of Glu (10.0%) and lowest amounts of Cys, Met and Trp (1.3%) (Table 3). Thus, the concept NPBCF is proposed, which is the PBCF normalized to the substrate AA composition, *i.e.*, the AA's peptide bond concentration. FNA had low PBCF of 1.5% for Met, well below the average of 5.0%, but when corrected to the low Met content in SPI, its NPBCF became 8.4%, well above the average of 5.0%, thus indicating FNA had good capability to hydrolyze Met at P1 in SPI. PBCF has also been earlier called Cleavage Frequency [29] or Preference Ratio [33] in porcine pepsin digestion of phosphorylase b and myoglobin, but in none of these studies was it normalized to substrate AA composition as done in the current study.

It should be noted that neither FNA nor pancreatin had high NPBCF for certain AA formed peptide bonds (*e.g.*, Cys, Asp, Glu and Pro). Peptides having these bonds can be hydrolyzed by pepsin and intestinal membrane anchored peptidases [31]. For instance, pepsin had a higher preference to Glu (Fig 2) [29, 33] compared to pancreatin (Figs 4 and 7). The observed low NPBCF values for Cys in all the three experiments were most likely because of all the three experiments being carried out in the absence of reducing reagents (such as 2-mercaptoethanol and dithiothreitol) to destruct the secondary structures of SPI. Under reducing conditions Cys in BSA substrate was well hydrolyzed by pancreatin and FNA, with a NPBCF of 9.5 and 3.5, respectively (our unpublished results). The low NPBCF values for Pro are because it is the only AA that forms peptide bond with a secondary amine and has a distinctive cyclic side chain which is known to be a structural disruptor in the middle of regular secondary structure elements such as α-helices and β-sheets. Beside the secondary structures that can be caused by Cys and Pro, N-glycosylation of Asn or phosphorylation of Ser and Tyr could also play a role in affecting peptide bond cleavage.

## Conclusions

This study was set up to characterize and compare two commercial exogenous feed proteases subtilisin-typed FNA and chymotrypsin-like NPP, endogenous pepsin, pancreatin and their

combinations in hydrolyzing a complex substrate of soy bean protein isolate (SPI) under three reaction conditions by nano LC-MS. The results obtained led to the proposal of NPBCF (Normalized Peptide Bond Cleavage Frequency) and PSBI (Protease Substrate Broadness Index). By NPBCF, the hydrolysis preference of a protease on different AA in a protein substrate at P1 can be characterized and compared without considering the bias due to the abundancy of this AA in the protein substrate. By the concept of PSBI, a general capability of a protease to hydrolyze any peptide bonds found in a protein substrate can be examined, which is a complementation to its activity assay that are usually based on initial hydrolysis rate at excess of substrate when the most preferred peptide bonds are first hydrolyzed. That FNA improved the PSBI of both pepsin and pancreatin under the three assay conditions is explained by its better complementarity with the endogenous proteases. Besides PSBI, a feed protease together with pancreatin generates higher amounts of smaller peptides from feed proteins could be favorable for absorption.

## Supporting information

**S1 Table. Primary LC-MS data listing of soy protein peptides generated in Experiment 1.** The peptides were generated at SPI:protease of 20:1 or 20:0.5:05 by pepsin, FNA, pepsin+FNA at pH3.7, and by pancreatin, FNA, and pancreatin+FNA at pH6.5. Information and parameters of the peptides listed are: UniProt accession number, monoisotopic m/z, calculated/theoretical mass, start and end residue number in candidate soy protein sequence, AA residue before the identified sequence, the identified peptide sequence (of which the last residue is P1), and AA residue after the identified sequence (P1).
(XLSX)

**S2 Table. Primary LC-MS data listing of soy protein peptides generated in Experiment 2.** The peptides were generated at SPI:protease of 100:1 or 100:0.5:05 by pepsin, FNA, pepsin +FNA at pH3.7, and by pancreatin, FNA, and pancreatin+FNA at pH6.5.
(XLSX)

**S3 Table. Primary LC-MS data listing of soy protein peptides generated in Experiment 3.** The peptides were generated at SPI:protease of 200:1 or 200:1:1 by pancreatin, FNA, NPP, pancreatin+FNA, and pancreatin+NPP at pH6.5.
(XLSX)

## Acknowledgments

The authors would like to thank M. Thorsen and S. Y. Bak from DuPont Advanced Analysis for their input to discussions on the technical aspects of the study, C. Evans, J. Kanarek, and H. Irving from DuPont Animal Health and Nutrition Management for their encouragement and support during the study. The authors also acknowledge Dr J. Buck (Reading, UK) for her assistance with the writing of this manuscript, which was sponsored by DuPont Nutrition & Biosciences, The Netherlands, in accordance with Good Publication Practice guidelines.

## Author Contributions

**Data curation:** Janne Bech Thoegersen.

**Investigation:** Janne Bech Thoegersen.

**Methodology:** Shukun Yu, Janne Bech Thoegersen.

**Project administration:** Shukun Yu, Karsten Mathias Kragh.

**Resources:** Shukun Yu, Karsten Mathias Kragh.

**Software:** Janne Bech Thoegersen.

**Supervision:** Karsten Mathias Kragh.

**Writing – original draft:** Shukun Yu.

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
