## [Decision Letter · Decision Letter 0]

12 Jun 2020

PONE-D-20-14957

Comparative study of protease hydrolysis reaction demonstrating Normalized Peptide Bond Cleavage Frequency and Protease Substrate Broadness Index

PLOS ONE

Dear Dr. Yu,

Thank you for submitting your manuscript to PLoS ONE. The paper has been reviewed by three experts in the field. All reviewers agree in that your paper deals with an interesting topic. However, they have different opinions about the robustness of your data and conclusions.  

*While reviewers no. 2 and 3 are rather positive and suggest doable revisions, reviewer no. 1 recommends rejection based on important limitations of your study, most notably that the protease *specificity typically cannot be described by only the P1 residue, and that the in vivo situation is much more complex that the model used: the brush border proteases and peptidases make a substantial contribution to the degradation of the food proteins. The first problem is also raised by reviewer no. 2 who requests revisions addressing this point. Another important problem of the manuscript is that it does not provide the primary data (MS analyses) as supplementary file. This important deficiency is pointed out by all three reviewers and should be addressed in your revised submission.* At any rate, after careful consideration, we consider that your manuscript needs major revisions. If you are prepared to undertake the work required, I would be pleased to reconsider my decision.*

We look forward to receiving your revised manuscript.

Kind regards,

Luis Menéndez-Arias, Ph. D.

Academic Editor

PLOS ONE

Journal Requirements:

2. We note that this submission reports a functional enzymological study with kinetic and thermodynamic data. The reporting of these data should include the temperature, pH and pressure, as well as the identity of the catalyst and its origins, the method of preparation, criteria for purity and assay conditions. We recommend that you refer to the Standards for Reporting Enzymology Data (STRENDA) of the Beilstein Institut for details regarding the adequate description of experimental conditions and reporting of enzyme activity data: https://www.beilstein-strenda-db.org/strenda/public/guidelines.xhtml. Please note that the Beilstein Institut’s STRENDA database automatically checks manuscript data for guideline compliance, as well as making them publicly available after publication and assigning them a specific DOI number for reference and tracking purposes. If you obtain a STRENDA Registry number (SRN) and PDF containing all your functional enzymology data, please include these as Supplementary files.

"The authors also acknowledge Dr Joelle Buck (Reading, UK) for her assistance with the writing of this manuscript, which was sponsored by DuPont Nutrition & Biosciences, The Netherlands, in accordance with 482 Good Publication Practice guidelines."

"The authors received no specific funding for this work."

Additionally, because some of your funding information pertains to [DuPont Nutrition & Biosciences], we ask you to provide an updated Competing Interests statement, declaring all sources of commercial funding.

In your Competing Interests statement, please confirm that your commercial funding does not alter your adherence to PLOS ONE Editorial policies and criteria by including the following statement: "This does not alter our adherence to PLOS ONE policies on sharing data and materials.” as detailed online in our guide for authors  http://journals.plos.org/plosone/s/competing-interests.  If this statement is not true and your adherence to PLOS policies on sharing data and materials is altered, please explain how.

4. Thank you for stating the following in the Financial Disclosure section:

"The authors received no specific funding for this work."

We note that one or more of the authors are employed by a commercial company: DuPont Nutrition & Biosciences.

4.1. Please provide an amended Funding Statement declaring this commercial affiliation, as well as a statement regarding the Role of Funders in your study. If the funding organization did not play a role in the study design, data collection and analysis, decision to publish, or preparation of the manuscript and only provided financial support in the form of authors' salaries and/or research materials, please review your statements relating to the author contributions, and ensure you have specifically and accurately indicated the role(s) that these authors had in your study. You can update author roles in the Author Contributions section of the online submission form.

4.2. Please also provide an updated Competing Interests Statement declaring this commercial affiliation along with any other relevant declarations relating to employment, consultancy, patents, products in development, or marketed products, etc. 

Reviewers' comments:

Reviewer's Responses to Questions

**Comments to the Author**

1. Is the manuscript technically sound, and do the data support the conclusions?

Reviewer #1: Partly

Reviewer #2: Partly

Reviewer #3: Yes

2. Has the statistical analysis been performed appropriately and rigorously? 

Reviewer #1: I Don't Know

Reviewer #2: Yes

Reviewer #3: Yes

3. Have the authors made all data underlying the findings in their manuscript fully available?

Reviewer #1: No

Reviewer #2: Yes

Reviewer #3: Yes

4. Is the manuscript presented in an intelligible fashion and written in standard English?

Reviewer #1: Yes

Reviewer #2: Yes

Reviewer #3: Yes

5. Review Comments to the Author

Reviewer #1: This manuscript by Yu et al., compares the proteolytic effect of various proteases on soy protein isolate, alone, or in combination, mimicking the pH conditions of stomach and small intestine. The authors propose the use of two parameters, Normalized Peptide Bond Cleavage Frequency, and Protease Substrate Broadness Index. They have concluded, that an exogenous protease FNA showed complementary effects with pepsin and pancreatin. There are several major limitations of this study, some of them also pointed out by the authors. (i) The protease specificity typically cannot be described by only the P1 residue, which is studied in this work. (ii) The in vivo situation is much more complex that the model used: the brush border proteases and peptidases make a substantial contribution to the degradation of the food proteins (iii) It is not determined that at the cleaved sites what was the conversion ratio - there is no data on the non-cleaved protein amounts.

The abstract states the comparison of two commercial proteases (FNA and NPP) with swine pepsin and pancreatin, however, only the the two latter ones are easily available commercially, form the manuscript the exact origin/NPP is not clear. Furthermore, the manuscript does not provide the primary data (MS analyses) as supplementary file.

Reviewer #2: The authors have studied the effects of exogenous peptidases which are added to animal feed to improve digestion of proteins. The peptidases examined are both from bacteria, and these are compared with porcine pepsin and a mixture of enzymes from pig pancreas known as 'pancreatin'. The authors demonstrate that addition of a subtilisin homologue from Bacillus amyloliquefaciens effectively increased digestion of soy protein isolate (an example of an animal feed), resulting in shorter peptides that could be more easily absorbed or digested by cells lining the intestine. The authors propose two new measures, Normalized Peptide Bond Cleavage Frequency and Protease Substrate Broadness Index, but applying these to a peptidase where specificity is not dominated by substrates binding at a single site would be a problem. Several other minor issues are detailed below.

Major point

As the authors themselves point out in the Discussion, not all peptidases have a preference solely for occupation of the P1 binding pocket. Of the 1,746 peptidases with known substrate cleavages in the MEROPS collection, 1,082 are known to bind only one or two different residues in P1 (62%). However, if peptidases where less than ten substrates are known are excluded, then only 113 out of a total of 517 peptidases have a preference of one or two residues in P1 (22%). The numbers of these 517 peptidases with substrate specificity directed to different binding pockets are: P4 97 (19%); P3 85 (16.5%); P2 81 (16%), P1' 44 (8.5%); P2' 68 (13%); P3' 78 (15%); P4' 86 (16.5%)). So the range is 13-22%.

Minor points

Throughout: my understanding is that pancreatin is a mix of enzymes, including lipases as well as peptidases. It is therefore incorrect to call pancreatin a 'protease'. This would be incorrect even if pancreatin was only a mix of peptidases.

page 3, line 60: replace 'faces' with 'faeces'.

page 4, line 89: replace 'a' with 'an' before 'S1 family member'.

page 5, lines 114-116: it isn't clear to me from this sentence whether the amounts of Glu, Gln, Asp and Asn were deduced from the genome sequencing data because it is difficult to distinguish between Glu and Gln or Asp and Asn, or whether only Glu, Gln, Asp and Asn were measured and the composition of other amino acids were estimated from them. I suspect the former, but please rephrase to sentence to make this clearer.

Table 1: My assumption is that soy protein isolate is a mix of proteins and that column A is a percentage, but neither is clearly explained in the text or the legend. Column B purports to be the number of each amino acid occupying the P1 substrate-binding site in peptidase FNA, but have the C-terminal residues of the proteins been excluded? Whether the protein is digested or not, its C-terminal amino acid could have been miscounted as a P1 residue; how much this affects the analysis will depend on the number of proteins in the mix. Finally, the bottom three rows are not aligned correctly with the rest of the table.

Table 2: Temperature and reaction time are the same in all rows. It would take less space simply to state in the legend that these were kept the same.

page 8, line 156: In a former career, I was a technician performing manual protein sequencing, and I used trifluoroacetic acid to break the N-terminal peptide bond. Please assure me that that is not happening here, because it would affect the results.

page 9, line 177: Why was the digestion time doubled in this experiment?

page 10, lines 197-198: I'm not sure whose benefit 'clarity' is for, here. It certainly makes the study simpler if substrate binding pockets other than P1 are ignored, but even trypsin shows some specifity for the P1' pocket (it can't accept Pro). Please refer to the major point above.

page 11, lines 221-222. I don't understand why the molecular masses of the peptidases affect the percentage of peptides generated by FNA in relation to pepsin.

page 11, line 223: replace "ration" with "ratio".

Fig. 6: I can't see a difference in the thickness of shade of the lines for SPI:FNA and SPI:Pancreatin+FNA. Please change one of them.

page 23, lines 384-394: please explain how PSBI would be calculated for a peptidase that showed preference in two substrate-binding pockets.

page 23, line 406: I would ask the authors to remove the statement "albeit to a lesser extent".

Finally, a question. Why not supplement feeds with amino acids? Wouldn't this achieve the same outcome?

Reviewer #3: Yu et al. from the company DuPont Nutrition & Biosciences investigate so-called feed proteases for farm animals, which can improve the efficient uptake of nutritional proteins, such as from soy beans. To this end, they assay the hydrolytic activity of several commercially available proteases and their combinations. Two interesting concepts, the Normalized Peptide Bond Cleavage Frequency and Protease Substrate Broadness Index, allow the analysis of their protein digest data after LC-MS. Thus, the whole study is straightforward and certainly relevant for the food industry. The abstract is a bit lengthy with too many technical details and contains too little information on results. Overall, there are not many flaws, while my only concern are the duplicate measurements in experiment 1. It would be more convincing to have consistently triplicates, but I am not sure, whether it is required in this type of experiment. The methodology and the description of the results are excellent and the discussion and conclusion are most appropriate. Therefore, the thorough correction of the following minor points would suffice to accept a revised version.

ABSTRACT

Lines 24-25: Basically, FNA is a mutant of subtilisin BPN´, information on NPP from Nocardiopsis prasina is hard to find, and pancreatin is not a well defined protease. The latter is an enzyme mixture that even contains amylases and lipases. These special facts should be explained for the general readership.

27: pH should be written throughout the manuscript with a space before the numbers.

INTRODUCTION

60: “faces“ is certainly feces.

77-78: Instead of “alkaline protease“ the term subtilisin should be used.

89: The EC number for NPP defines it as S1 family protease, perhaps trypsin-like, chymotrypsin- or elastase-like. This information should be specified here and in the abstract.

99-101: References for the definitions of NPBCF, PBCF and PSBI are missing.

MATERIALS AND METHODS

107-108: The problem of the mixture pancreatin is obviously that neither units nor defined concentration in mg/ml for a single protease can be given. At least the percentage of the major components should be given here.

110-111: “random acting serine protease“ does not sound very scientific, I prefer unspecific serine protease.

118-121: I consider this table more as a result than a method description and it seems misplaced. PSBI is missing.

136: The definition of pancreatin as protease is flawed, since it is an enzyme mixture containing amylases and lipases.

142: It would be more logical to describe “In vitro protease digestion of SPI“ before the previous section.

145: Duplicates in experiment 1 could be acceptable, however, it would be more consistent with general enzymatic studies and the rest of the study itself to have triplicates. This blemish could be eliminated by an additional measurement series.

151-153: The reaction volumes do not correspond to the ones in table 2, which contains no 200 μl samples.

148: While I see no problem for denatured SPI samples with 8 M urea for in vitro experiments, these conditions seem quite artificial. In addition, some of the proteins might refold in the pH 3.7 and pH 6.5 buffers. Otherwise, pepsin digest in the stomach will generate largely clipped three-dimensional folded proteins and shorter polypeptides. These points should be addressed in the discussion.

173-176: The reaction volume here is 175 μl, whereas table 2 shows 25 μl.

196-197: I prefer scissile bond instead of “cleft bond“, as the Schechter-Berger nomenclature is valid for intact substrates. The standard usage is P4-P1-P1'-P3' (or P4'), meaning that the prime symbol is used, not an apostrophe.

199-200: Apparently, table 1 belongs rather to this section.

RESULTS

218: Replace “alkaline protease“ with subtilisin.

223: “ration“ is certainly ratio.

313-315: The curves for FNA and FNA+pancreatin digests cannot really be distinguished. Another type for the lines should be used, perhaps grey and dotted.

330: “Sprotease“ is certainly a typo.

331: The low NPBCF for Cys is remarkable, as well as for Pro, and the acidic residues. As for Cys this should be discussed for all residues.

344-345 I guess p.p. is not a standard abbreviation, so use % here and in the following sections, if that is correct.

DISCUSSION

382: P1-P1' requires the prime symbol.

384: The URL for MEROPS is acceptable, however, the site gives a recent citation: Rawlings, N.D., Barrett, A.J., Thomas, P.D., Huang, X., Bateman, A. & Finn, R.D. (2018) The MEROPS database of proteolytic enzymes, their substrates and inhibitors in 2017 and a comparison with peptidases in the PANTHER database. Nucleic Acids Res 46, D624-D632.

406-407: Should be P4-P2.

425: Approximately has to be written out.

427: Correct to “which are transported“ or “which are being transported“.

431: Change to the correct nomenclature.

437-438: A short discussion of the most relevant posttranslational modifications, such as disulfide formation (is done), N-glycosylation of Asn or phosphorylation of Ser and Tyr could demonstrate that other factors play a role in such studies, also for SPI.

CONCLUSIONS

468: Change to “a general capability of a protease“ and delete “AA formed“.

6. PLOS authors have the option to publish the peer review history of their article (what does this mean?). If published, this will include your full peer review and any attached files.

Reviewer #1: No

Reviewer #2: No

Reviewer #3: No

---

## [Author Response · Author response to Decision Letter 0]

4 Aug 2020

To: Dr Luis Menéndez-Arias, Academic Editor Plos One 

Aarhus, Denmark, July 02, 2020

Dear Dr Luis Menéndez-Arias, 

We would like to thank you and the three reviewers for the highly valuable suggestions for revising our ms PONE-D-20-14957. We accept all these without any reservation and have revised it. As suggested, all the Primary LC-MS data as Supporting information S1, S2, and S3 Tables (for Experiment 1, 2, and 3) are now provided. For the explanation to the Reviewers’ points and our revisions, please you may go to Page 5 below.

Two new notions are proposed in the current study. It is noticeable that none of the Reviewers had major opinions on the new notion of Normalized Peptide Bond Cleavage Frequency (NPBCF) (Fig 1-8A), while Reviewer 1 and 2 had comments on the second new notion, viz, Protease Substrate Broadness Index (PSBI) (Fig 8B), which is the reciprocal of Standard deviation of mean NPBCF shown in Fig 8A. We have now addressed their comments in both Result Section and Discussion section. These also include an additional experiment which was performed with sequencing grade Endoproteinase Glu-C to give an example what the PSBI would be for a more specific protease than the unspecific proteases (pepsin, subtilisin-typed FNA, chymotrypsin-like NPP etc) examined in the study.

Concerning funding, we would also like to change “The authors received no specific funding for this work." To: This work was funded by “E. I. du Pont de Nemours and Company”. This could be more appropriate as we are DuPont employees.

Sincerely yours, Shukun Yu, PhD, Senior staff scientist, DuPont Nutrition & Biosciences, Edwin Rahrs Vej 38, DK 8220 Brabrand, Aarhus, Denmark, Tel: 0045 8943 5495; Cell: +45 24675 165

PS. Your email on our ms with the reviewers’ points:

From: em.pone.0.6be764.8f0763c8@editorialmanager.com <em.pone.0.6be764.8f0763c8@editorialmanager.com> On Behalf Of PLOS ONE

Sent: 12. juni 2020 11:18

To: Shukun Yu <Shukun.Yu@dupont.com>

Subject: [EXTERNAL] PLOS ONE Decision: Revision required [PONE-D-20-14957] - [EMID:03dbe36808100376]

PONE-D-20-14957

Comparative study of protease hydrolysis reaction demonstrating Normalized Peptide Bond Cleavage Frequency and Protease Substrate Broadness Index

PLOS ONE

Dear Dr. Yu,

Thank you for submitting your manuscript to PLoS ONE. The paper has been reviewed by three experts in the field. All reviewers agree in that your paper deals with an interesting topic. However, they have different opinions about the robustness of your data and conclusions. 

While reviewers no. 2 and 3 are rather positive and suggest doable revisions, reviewer no. 1 recommends rejection based on important limitations of your study, most notably that the protease specificity typically cannot be described by only the P1 residue, and that the in vivo situation is much more complex that the model used: the brush border proteases and peptidases make a substantial contribution to the degradation of the food proteins. The first problem is also raised by reviewer no. 2 who requests revisions addressing this point. Another important problem of the manuscript is that it does not provide the primary data (MS analyses) as supplementary file. This important deficiency is pointed out by all three reviewers and should be addressed in your revised submission. At any rate, after careful consideration, we consider that your manuscript needs major revisions. If you are prepared to undertake the work required, I would be pleased to reconsider my decision.

We look forward to receiving your revised manuscript.

Kind regards,

Luis Menéndez-Arias, Ph. D.

Academic Editor

PLOS ONE

Journal Requirements:

2. We note that this submission reports a functional enzymological study with kinetic and thermodynamic data. The reporting of these data should include the temperature, pH and pressure, as well as the identity of the catalyst and its origins, the method of preparation, criteria for purity and assay conditions. We recommend that you refer to the Standards for Reporting Enzymology Data (STRENDA) of the Beilstein Institut for details regarding the adequate description of experimental conditions and reporting of enzyme activity data: https://www.beilstein-strenda-db.org/strenda/public/guidelines.xhtml. Please note that the Beilstein Institut’s STRENDA database automatically checks manuscript data for guideline compliance, as well as making them publicly available after publication and assigning them a specific DOI number for reference and tracking purposes. If you obtain a STRENDA Registry number (SRN) and PDF containing all your functional enzymology data, please include these as Supplementary files.

"The authors also acknowledge Dr Joelle Buck (Reading, UK) for her assistance with the writing of this manuscript, which was sponsored by DuPont Nutrition & Biosciences, The Netherlands, in accordance with 482 Good Publication Practice guidelines."

"The authors received no specific funding for this work."

Additionally, because some of your funding information pertains to [DuPont Nutrition & Biosciences], we ask you to provide an updated Competing Interests statement, declaring all sources of commercial funding.

In your Competing Interests statement, please confirm that your commercial funding does not alter your adherence to PLOS ONE Editorial policies and criteria by including the following statement: "This does not alter our adherence to PLOS ONE policies on sharing data and materials.” as detailed online in our guide for authors http://journals.plos.org/plosone/s/competing-interests. If this statement is not true and your adherence to PLOS policies on sharing data and materials is altered, please explain how.

4. Thank you for stating the following in the Financial Disclosure section:

"The authors received no specific funding for this work."

We note that one or more of the authors are employed by a commercial company: DuPont Nutrition & Biosciences.

4.1. Please provide an amended Funding Statement declaring this commercial affiliation, as well as a statement regarding the Role of Funders in your study. If the funding organization did not play a role in the study design, data collection and analysis, decision to publish, or preparation of the manuscript and only provided financial support in the form of authors' salaries and/or research materials, please review your statements relating to the author contributions, and ensure you have specifically and accurately indicated the role(s) that these authors had in your study. You can update author roles in the Author Contributions section of the online submission form.

4.2. Please also provide an updated Competing Interests Statement declaring this commercial affiliation along with any other relevant declarations relating to employment, consultancy, patents, products in development, or marketed products, etc. 

Reviewers' comments:

Reviewer's Responses to Questions

Comments to the Author

1. Is the manuscript technically sound, and do the data support the conclusions?

Reviewer #1: Partly. Our answers: In this study we have 11 treatments consisting of single protease and protease combinations (Pepsin, FNA, pepsin+FNA at pH3.7 etc……). Data from all these treatments supported our notions of Normalized Peptide Bond Cleavage Frequency (Fig 1-8A) and Protease Substrate Broadness Index (PSBI) (Fig 8B). Now primary LC-MS data are provided as Support information S1, S2, and S3 Tables.

Reviewer #2: Partly. Our answer: see above

Reviewer #3: Yes. Our answer: 😊

2. Has the statistical analysis been performed appropriately and rigorously? 

Reviewer #1: I Don't Know. Our answer: the calculation of PSBI from all these 11 treatments is based on the stdev of the number of peptides generated (Table 3 is given as an example), and PSBI is the reciprocal of Stdev of mean NPBCF.

Reviewer #2: Yes. Our answer: 😊

Reviewer #3: Yes. Our answer: 😊

3. Have the authors made all data underlying the findings in their manuscript fully available?

Reviewer #1: No. Our answer: We have 12 figures and one Table in the manuscripts to demonstrate these two novel notions. Now primary LC-MS data are now provided as Support information S1, S2 and S3 Tables.

Reviewer #2: Yes. Our answer: 😊

Reviewer #3: Yes. Our answer: 😊

4. Is the manuscript presented in an intelligible fashion and written in standard English?

Reviewer #1: Yes. Our answer: 😊

Reviewer #2: Yes. Our answer: 😊

Reviewer #3: Yes. Our answer: 😊

5. Review Comments to the Author

Reviewer #1: This manuscript by Yu et al., compares the proteolytic effect of various proteases on soy protein isolate, alone, or in combination, mimicking the pH conditions of stomach and small intestine. The authors propose the use of two parameters, Normalized Peptide Bond Cleavage Frequency, and Protease Substrate Broadness Index. They have concluded, that an exogenous protease FNA showed complementary effects with pepsin and pancreatin. There are several major limitations of this study, some of them also pointed out by the authors. (i) The protease specificity typically cannot be described by only the P1 residue, which is studied in this work. Our answers: Please see our answers to “Major point” by Reviewer 2 in detail below. (ii) The in vivo situation is much more complex that the model used: the brush border proteases and peptidases make a substantial contribution to the degradation of the food proteins. Our answers: Agree and we did not in any way state the current in vitro study would replace animal protein digestion trials. Indeed, we have discussed intestinal membrane anchored peptidases in processing peptides with 10 or less AA residues with the citation of two references (27-28); (iii) It is not determined that at the cleaved sites what was the conversion ratio - there is no data on the non-cleaved protein amounts. Our answers: No, this is not a Degree of Protein Hydrolysis study (such studies on these industrial enzymes, pepsin and pancreatin could be found in product brochures and published scientific papers). The data obtained in all these 3 experiments meet the need to propose the two novel concepts, viz, NPBCF and PSBI.

The abstract states the comparison of two commercial proteases (FNA and NPP) with swine pepsin and pancreatin, however, only the the two latter ones are easily available commercially, form the manuscript the exact origin/NPP is not clear. Our answers: Two additional references on the chymotrypsin-like NPP are now provided. Furthermore, the manuscript does not provide the primary data (MS analyses) as supplementary file. Our answers: The primary MS data for all the 3 experiments are now provided as Support Information S1, S2, and S3 Tables.

Reviewer #2: The authors have studied the effects of exogenous peptidases which are added to animal feed to improve digestion of proteins. The peptidases examined are both from bacteria, and these are compared with porcine pepsin and a mixture of enzymes from pig pancreas known as 'pancreatin'. The authors demonstrate that addition of a subtilisin homologue from Bacillus amyloliquefaciens effectively increased digestion of soy protein isolate (an example of an animal feed), resulting in shorter peptides that could be more easily absorbed or digested by cells lining the intestine. The authors propose two new measures, Normalized Peptide Bond Cleavage Frequency and Protease Substrate Broadness Index, but applying these to a peptidase where specificity is not dominated by substrates binding at a single site would be a problem. Several other minor issues are detailed below.

Major point

As the authors themselves point out in the Discussion, not all peptidases have a preference solely for occupation of the P1 binding pocket. Of the 1,746 peptidases with known substrate cleavages in the MEROPS collection, 1,082 are known to bind only one or two different residues in P1 (62%). However, if peptidases where less than ten substrates are known are excluded, then only 113 out of a total of 517 peptidases have a preference of one or two residues in P1 (22%). The numbers of these 517 peptidases with substrate specificity directed to different binding pockets are: P4 97 (19%); P3 85 (16.5%); P2 81 (16%), P1' 44 (8.5%); P2' 68 (13%); P3' 78 (15%); P4' 86 (16.5%)). So the range is 13-22%.

Our answers: What said above by Reviewer #2 is indeed true: only 113 out of a total of 517 peptidases have a preference of one or two residues in P1 (13-22%). However, among the vast number of proteases found in nature, only dozens of proteases and variants derived thereof are used in the industries (laundry, grain processing, feed and food industries) in the last six decades. These industry proteases have been and still are dominantly subtilisins and their variants, followed by bacterial metal peptidases and to a much lesser extent plant extracted proteases (e.g., papain). Understandably an important feature of industrial proteases is their lower substrate specificity enabling them to generate a high Degree of Hydrolysis of their substrates. These have now been addressed in both Results section and Discussion section. An important observation in the study, which has now been underlined in the revised version, is that, even though PSBI’s calculation is based on the number of peptides generated having the same AA at C-terminal (i.e., the same AA at P1), the number of peptides generated from SPI (soy protein isolate) shown in Figure 1 and 5 were unavoidably affected by the residues of P1-P4 and P1'-P4', by the post translational modifications and secondary structures to these residues. That is, the peptides generated from SPI by these proteases were a result of the interactions between protease’s pockets S4-S1 and S1ꞌ-S4ꞌ with P4-P1 and P1ꞌ-P4ꞌ AA residues in the substrate SPI. In the context of the Reviewer’s point we have now made it more unambiguous for the application area of PSBI, i.e., 1) the substrate for the estimation of PSBI should be proteins having good representations of all 20 AAs; 2) it is mainly for industrial proteases having less preference to AA residues on either side of the scissile bond. We have explained this in the Discussion section why we have chosen soy protein for the current study to estimate the PSBI of the proteases instead of myoglobin and BSA reported in the literature and which we also examined. Furthermore, we have performed an additional experiment to examine the NPBCF and PSBI of the more specific Endoproteinase Glu-C used in protein sequencing, which showed 15 times higher preference (NPBCF) for Glu than for Asp at P1 and a PSBI value two times lower than pancreatin.

In the Result Section, the revised texts are: …………. Table 3 further shows the calculation of the parameters of NPBCF and PSBI. Thus, in Column B the first number of 94 means average of 94 peptides with different lengths but all with Ala (P1) at their C-termini were detected under the experiment setup of Experiment 1. The generation of the 94 peptides all with Ala at P1 also indicates FNA had less selectivity on the AAs residues in SPI flanking the P1 of Ala, i.e., P4-P2, P1ꞌ-P4ꞌ, whose identities can be found in S1 Tables. …………

In the Discussion Section, the revised texts are: In defining PSBI even though the data on the AAs at position P1 of the scissile bond in SPI are weighed, it should be mentioned that the AAs positioned at P4-P2 and P1'-P4' which can be found in S1-3 Tables are not neglected. This is because complex substrate of SPI had been used and the number of peptides generated (Fig 1, 5) by each of the proteases and protease combinations from SPI was unavoidably affected by AAs on either side of the scissile bond and even by the presence of secondary structures like disulfide bonds these AAs were involved in. The concept of the PSBI could be primarily useful for the assessment of animal feed proteases, detergent proteases, and proteases used in the hydrolysis industries, i.e., industrial proteases that a lower bias toward AAs near the scissile bonds is desirable. It can be envisaged that more specific proteases will have lower PSBI values and under the conditions of Experiment 3 Endoproteinase Glu-C from S. aureus strain V8 which favours Glu at P1 generated 4 times less peptides than pancreatin and had a PSBI value of 11.0, two times lower than pancreatin. It may also be stressed that substrates to be used for the estimation of PSBI should be proteins having known AA sequences preferably with all 20 AAs well presented. Synthetic substrates like Succinyl-AAPX-pNA which has been used in determining the most favourable AA at P1 for NPP homologue [22] is not suitable. 

Minor points

Throughout: my understanding is that pancreatin is a mix of enzymes, including lipases as well as peptidases. It is therefore incorrect to call pancreatin a 'protease'. This would be incorrect even if pancreatin was only a mix of peptidases. Our answers: Agree, and now it is explained in the Abstract concisely. The exact amount of protease, amylase and lipase in the batch of pancreatin used are given under Material and Method Section.

page 3, line 60: replace 'faces' with 'faeces'. Our answer: Done

page 4, line 89: replace 'a' with 'an' before 'S1 family member'. Our answer: Done

page 5, lines 114-116: it isn't clear to me from this sentence whether the amounts of Glu, Gln, Asp and Asn were deduced from the genome sequencing data because it is difficult to distinguish between Glu and Gln or Asp and Asn, or whether only Glu, Gln, Asp and Asn were measured and the composition of other amino acids were estimated from them. I suspect the former, but please rephrase to sentence to make this clearer. Our answer: Yes, it is the former. It is now rephrased: the amounts of Glu, Gln, Asp and Asn were deduced from the amounts of Glu+Gln and Asp+Asn provided by the vendor and the Glu/Gln, Asp/Asn ratio found in genome sequencing data of soy protein.

Table 1: My assumption is that soy protein isolate is a mix of proteins and that column A is a percentage, but neither is clearly explained in the text or the legend (Our answer: yes, now % is added, short explanation is given). Column B purports to be the number of each amino acid occupying the P1 substrate-binding site in peptidase FNA, but have the C-terminal residues of the proteins been excluded? Whether the protein is digested or not, its C-terminal amino acid could have been miscounted as a P1 residue; how much this affects the analysis will depend on the number of proteins in the mix (Our answer: In order to give a better description of this Table, it is now moved to Result Section as Table 3 suggested by Reviewer #3 below. For example, the title of Column A is now revised to get a better understanding: with the first listed Ala as example in the Table, it indicates 94 peptides though with different sequences and lengths but all with Ala as their C-termini (i.e., Ala as P1 occupying S1 binding pocket of FNA) were detected. The exact AA sequences for these 94 peptides are found in Support Information S1-S3 Tables where AAs flanking the identified sequences (N-, C-terminal residues) are given. If more explanation is needed, please referrer to “The Major point” above). Finally, the bottom three rows are not aligned correctly with the rest of the table (Our answer: now adjusted).

Table 2: Temperature and reaction time are the same in all rows. It would take less space simply to state in the legend that these were kept the same. (Our answer: these two columns are now deleted).

page 8, line 156: In a former career, I was a technician performing manual protein sequencing, and I used trifluoroacetic acid to break the N-terminal peptide bond. Please assure me that that is not happening here, because it would affect the results. Our answers: Yes, concentrated TFA has been used to cleave certain peptide bonds in either gas or liquid phase and the reaction takes usually several days. We have used 0.1% TFA in our HPLC part of LC-MS and 0.1% TFA is also commonly used in HPLC for separating peptides in decades. Such low concentration and short HPLC run time will probably never cause any peptide bond cleavages.

page 9, line 177: Why was the digestion time doubled in this experiment? Our answers: We wanted to have varied reaction conditions including reaction time (also SPI to protease ratios, instrument, use of denaturants etc) and under such varied reaction conditions the same conclusions could still be obtained, Yes, it did, i.e., in all the 3 experiments with varied conditions, the conclusions are the same as summarized in the Conclusion section.

page 10, lines 197-198: I'm not sure whose benefit 'clarity' is for, here. It certainly makes the study simpler if substrate binding pockets other than P1 are ignored, but even trypsin shows some specifity for the P1' pocket (it can't accept Pro). Please refer to the major point above. Our answers: The sentence “For clarity AAs at P4-P2 and P1'-P3' were not considered in the current study” is now deleted. Please refer to answers to the Major Point above. For the proteases tested, all 20 AAs were accepted at P1ꞌ including Pro (S1-3 Tables).

page 11, lines 221-222. I don't understand why the molecular masses of the peptidases affect the percentage of peptides generated by FNA in relation to pepsin. Our answers: Yes, it did, but it makes no meaning to describe it both on weight and on mol mass basis. Now it is described only on weight basis.

page 11, line 223: replace "ration" with "ratio". Our answers: Done.

Fig. 6: I can't see a difference in the thickness of shade of the lines for SPI:FNA and SPI:Pancreatin+FNA. Please change one of them. Our answer: Good point and now changed.

page 23, lines 384-394: please explain how PSBI would be calculated for a peptidase that showed preference in two substrate-binding pockets. Our answers: The procedure is the same for proteases exemplified in the current study. The protease that has preference on both P1 and P1' can have a low PSBI. For this question, the PSBI for Glu-C, a highly specific endopeptidase for Glu as P1 is included, which is approximately 3 times lower than FNA. 

page 23, line 406: I would ask the authors to remove the statement "albeit to a lesser extent". Our answers: Done.

Finally, a question. Why not supplement feeds with amino acids? Wouldn't this achieve the same outcome? Our answer: We healthy humans normally eat proteins 1g per kilo body weight per day; we only drink or get injections of amino acid mixture if our digestion system is jeopardized. Amino acids taste badly beside its high price. However, we do add Lysine and Methionine as feed additives. All these are perhaps irrelevant in the context of the current study.

Reviewer #3: Yu et al. from the company DuPont Nutrition & Biosciences investigate so-called feed proteases for farm animals, which can improve the efficient uptake of nutritional proteins, such as from soy beans. To this end, they assay the hydrolytic activity of several commercially available proteases and their combinations. Two interesting concepts, the Normalized Peptide Bond Cleavage Frequency and Protease Substrate Broadness Index, allow the analysis of their protein digest data after LC-MS. Thus, the whole study is straightforward and certainly relevant for the food industry. The abstract is a bit lengthy with too many technical details and contains too little information on results. Our answer: the abstract is shortened while giving room for what FNA and NPP are. Overall, there are not many flaws, while my only concern are the duplicate measurements in experiment 1. It would be more convincing to have consistently triplicates, but I am not sure, whether it is required in this type of experiment. The methodology and the description of the results are excellent and the discussion and conclusion are most appropriate. Therefore, the thorough correction of the following minor points would suffice to accept a revised version. Our answer: Experiment 1 was done in duplicated but very good reproducibility was achieved (see the stdev in Fig 1 and Table 3).

ABSTRACT

Lines 24-25: Basically, FNA is a mutant of subtilisin BPN´, information on NPP from Nocardiopsis prasina is hard to find, and pancreatin is not a well defined protease. The latter is an enzyme mixture that even contains amylases and lipases. These special facts should be explained for the general readership. Our answer: Very good suggestion and now done properly in Abstract, Introduction and Material method section.

27: pH should be written throughout the manuscript with a space before the numbers. Done

INTRODUCTION

60: “faces“ is certainly feces. Our answer: Done

77-78: Instead of “alkaline protease“ the term subtilisin should be used. Our answer: Done

89: The EC number for NPP defines it as S1 family protease, perhaps trypsin-like, chymotrypsin- or elastase-like. This information should be specified here and in the abstract. Our answer: Done at both places now. NPP is chymotrypsin-like S1 family member endopeptidases.

99-101: References for the definitions of NPBCF, PBCF and PSBI are missing. Our answer: These two concepts are first time proposed by the authors here and their discussions with what are reported in the literature (26. Ahn et al. 2013; 30. Zhang et al 2008, both references defined PBCF) have already been discussed in the Discussion Section.

MATERIALS AND METHODS

107-108: The problem of the mixture pancreatin is obviously that neither units nor defined concentration in mg/ml for a single protease can be given. At least the percentage of the major components should be given here. Our answer: Agreed and the major components are given with also the amount of proteases in it (units/mg pancreatin) for the batch used.

110-111: “random acting serine protease“ does not sound very scientific, I prefer unspecific serine protease. Our answer: Done

118-121: I consider this table more as a result than a method description and it seems misplaced. PSBI is missing. Our answer: Agreed completely and Table 1 is replaced under Result section as Table 3. There is now a description on this Table including the calculation of PSBI.

136: The definition of pancreatin as protease is flawed, since it is an enzyme mixture containing amylases and lipases. Our answer: Agreed and corrected by adding amylase and lipase and amount of protease for the pancreatin batch used is given also.

142: It would be more logical to describe “In vitro protease digestion of SPI“ before the previous section. Our answer: Agree and this has been done now.

145: Duplicates in experiment 1 could be acceptable, however, it would be more consistent with general enzymatic studies and the rest of the study itself to have triplicates. This blemish could be eliminated by an additional measurement series. Our answer: From Fig 1 (and Table 3) one can see the Stedev of duplicated trial under the conditions of Experiment is at least as good as the triplicated trial of Experiment 2. Apparently the Stdev is larger when substrate SPI:protease is higher. The number of repetitions for protease digestion have been discussed already in the Discussion Section by having the Reference 26 of Ahn J, Cao M-J, Yu YQ, Engen JR, 2013 but supplemented now with a couple more sentences.

151-153: The reaction volumes do not correspond to the ones in table 2, which contains no 200 μl samples. Our answer: the 10kd solution-free spinfilter was transferred to a new spinfilter tube, not the reaction mixture. Now this has been rephrased.

148: While I see no problem for denatured SPI samples with 8 M urea for in vitro experiments, these conditions seem quite artificial. In addition, some of the proteins might refold in the pH 3.7 and pH 6.5 buffers. Otherwise, pepsin digest in the stomach will generate largely clipped three-dimensional folded proteins and shorter polypeptides. These points should be addressed in the discussion. Our answer: Yes, good points and now done as suggested, namely urea was used in Experiment 1-2 and SDS was used in experiment 3 in order to have different conditions to see if the results obtained shall support the two notions proposed.

173-176: The reaction volume here is 175 μl, whereas table 2 shows 25 μl. Our answer: the S-trap is solvent free since the 150ul buffer used to wash the S-trap has passed through during the centrifugation step (the same as questions to 151-153 above).

196-197: I prefer scissile bond instead of “cleft bond“, as the Schechter-Berger nomenclature is valid for intact substrates. The standard usage is P4-P1-P1'-P3' (or P4'), meaning that the prime symbol is used, not an apostrophe. Our answer: Agree and corrected.

199-200: Apparently, table 1 belongs rather to this section. Our answer: Agree and Done as suggested above also.

RESULTS

218: Replace “alkaline protease“ with subtilisin. Our answer: Done

223: “ration“ is certainly ratio. Our answer: Corrected

313-315: The curves for FNA and FNA+pancreatin digests cannot really be distinguished. Another type for the lines should be used, perhaps grey and dotted. Our answer: Done (double lines were chosen)

330: “Sprotease“ is certainly a typo. Our answer: Corrected

331: The low NPBCF for Cys is remarkable, as well as for Pro, and the acidic residues. As for Cys this should be discussed for all residues. Our answer: These are now mentioned in the Results section and have already been discussed thoroughly in the Discussion section, e.g. we indicated if reducing agent was present Cys at P1 was well hydrolyzed. We made it now even more clear, also for Pro at P1. Both Cys and Pro residues affect proteins’ secondary structures.

344-345 I guess p.p. is not a standard abbreviation, so use % here and in the following sections, if that is correct. Our answer: “P.p”. is replaced with “percentage point”

DISCUSSION

382: P1-P1' requires the prime symbol. Our answer: Done

384: The URL for MEROPS is acceptable, however, the site gives a recent citation: Rawlings, N.D., Barrett, A.J., Thomas, P.D., Huang, X., Bateman, A. & Finn, R.D. (2018) The MEROPS database of proteolytic enzymes, their substrates and inhibitors in 2017 and a comparison with peptidases in the PANTHER database. Nucleic Acids Res 46, D624-D632. Our answer: Both references have been cited already (see reference 19).

406-407: Should be P4-P2. Our answer: Done

425: Approximately has to be written out. Our answer: Done

427: Correct to “which are transported“ or “which are being transported“. Our answer: Done

431: Change to the correct nomenclature. Our answer: Done

437-438: A short discussion of the most relevant posttranslational modifications, such as disulfide formation (is done), N-glycosylation of Asn or phosphorylation of Ser and Tyr could demonstrate that other factors play a role in such studies, also for SPI. Our answer: Done in the context also proline that affects the secondary structures

CONCLUSIONS

468: Change to “a general capability of a protease“ and delete “AA formed“. Our answer: Done

6. PLOS authors have the option to publish the peer review history of their article (what does this mean?). If published, this will include your full peer review and any attached files.

Do you want your identity to be public for this peer review? For information about this choice, including consent withdrawal, please see our Privacy Policy.

Reviewer #1: No

Reviewer #2: No

Reviewer #3: No

---

## [Decision Letter · Decision Letter 1]

26 Aug 2020

PONE-D-20-14957R1

Comparative study of protease hydrolysis reaction demonstrating Normalized Peptide Bond Cleavage Frequency and Protease Substrate Broadness Index

PLOS ONE

Dear Dr. Yu,

Thank you for submitting your revised manuscript to PLoS ONE. After this round of revision, reviewers think that the manuscript has been improved and most queries and questions have been satisfactorily addressed. However, there are still some minor corrections to be done before the manuscript is accepted for publication (see reports below).

Although your manuscript needs revision, we believe that this is minor and the article should be publishable after appropriate changes have been done. If you are prepared to undertake the work required, I would be pleased to reconsider my decision.

We look forward to receiving your revised manuscript.

Kind regards,

Luis Menéndez-Arias, Ph. D.

Academic Editor

PLOS ONE

Reviewers' comments:

Reviewer's Responses to Questions

**Comments to the Author**

1. If the authors have adequately addressed your comments raised in a previous round of review and you feel that this manuscript is now acceptable for publication, you may indicate that here to bypass the “Comments to the Author” section, enter your conflict of interest statement in the “Confidential to Editor” section, and submit your "Accept" recommendation.

Reviewer #1: All comments have been addressed

Reviewer #2: All comments have been addressed

Reviewer #3: All comments have been addressed

2. Is the manuscript technically sound, and do the data support the conclusions?

Reviewer #1: Yes

Reviewer #2: Yes

Reviewer #3: Yes

3. Has the statistical analysis been performed appropriately and rigorously? 

Reviewer #1: Yes

Reviewer #2: Yes

Reviewer #3: Yes

4. Have the authors made all data underlying the findings in their manuscript fully available?

Reviewer #1: Yes

Reviewer #2: Yes

Reviewer #3: Yes

5. Is the manuscript presented in an intelligible fashion and written in standard English?

Reviewer #1: Yes

Reviewer #2: Yes

Reviewer #3: Yes

6. Review Comments to the Author

Reviewer #1: (No Response)

Reviewer #2: The authors have made the changes suggested by myself and the other reviewers and the paper is not acceptable for publication provided the following minor points, mostly errors in grammar introduced during the revision, are addressed.

Line 428: Remove "that" from "This is because that a protein".

Line 429: It would be better to say that "each peptide bond can be regarded as a potential substrate".

Line 444: insert "out" after "carried".

Line 483: It is usual for numbers under ten to be spelt out in full; so replace "4" with "four".

Line 486: replace "presented" with "represented".

Line 487-488: There is no need to capitalize "succinyl". Succinyl-AAPX-pNA is not a single substrate but range of substrates depending on the amino acid in the P1 position (shown as "X"), so replace "has" with "have" and "is" with "are". I suggest putting commas around the clause "which has been used ... at P1".

Line 497: "triplicate" should be plural.

Line 499: Replace "Beside" with "Besides".

Line 512: replace "present" with "represent".

lines 513-515: Considering that a prerequisite of the experimental procedure is the denaturation of SPI with urea prior to digestion, it seems unlikely that much tertiary structure of disulfide bridges remain. So this negates one of the advantages of SPI over the oxidized insulin B-chain.

Table 3. "Tryptophan" is mis-spelt.

Reviewer #3: The authors Yu, Thoergersen and Kragh have done a very good job in revising their manuscript. I think no further explanation of my answers to the above questions is required. As I read all their responses to all three reviewers comments, it is clear that they meticulously corrected every critical point. Thus, I can only suggest immediate acception of the study and would like to congratulate the authors for their work. It would be interesting to see, whether they can further develop the two novel concepts of NPBCF and PSBI beyond the food protein context. In fact, I see some potential for other applications in the protease field.

7. PLOS authors have the option to publish the peer review history of their article (what does this mean?). If published, this will include your full peer review and any attached files.

Reviewer #1: No

Reviewer #2: No

Reviewer #3: No

---

## [Author Response · Author response to Decision Letter 1]

27 Aug 2020

Reviewer 1 has no further comments, 

Reviewer 3 wanted to correct the typo of trptophan to tryptophan (done). 

Reviewer 2 has some suggestions on the wording and spelling. All of them are accepted and corrected (see below). 

Reviewer #2: The authors have made the changes suggested by myself and the other reviewers and the paper is not acceptable for publication provided the following minor points, mostly errors in grammar introduced during the revision, are addressed.

Line 428: Remove "that" from "This is because that a protein"._Done

Line 429: It would be better to say that "each peptide bond can be regarded as a potential substrate"._Done

Line 444: insert "out" after "carried"._Done

Line 483: It is usual for numbers under ten to be spelt out in full; so replace "4" with "four". _Done

Line 486: replace "presented" with "represented"._Done

Line 487-488: There is no need to capitalize "succinyl". Succinyl-AAPX-pNA is not a single substrate but range of substrates depending on the amino acid in the P1 position (shown as "X"), so replace "has" with "have" and "is" with "are". I suggest putting commas around the clause "which has been used ... at P1"._Done

Line 497: "triplicate" should be plural. _Done

Line 499: Replace "Beside" with "Besides"._Done

Line 512: replace "present" with "represent"._Done

lines 513-515: Considering that a prerequisite of the experimental procedure is the denaturation of SPI with urea prior to digestion, it seems unlikely that much tertiary structure of disulfide bridges remain. So this negates one of the advantages of SPI over the oxidized insulin B-chain. _the word tertiary is deleted.

We also made minor improvements of Fig 1 (added 20:0.5:0.5, 100:0.5:0.5), Fig 3A (deleted a space), Fig 8A and 8B (corrected typos). All these figures were saved as TIF and converted at https://pacev2.apexcovantage.com

Bst rgs, Dr Shukun Yu, Senior staff scientist, DuPont Nutrition & Biosciences, Edwin Rahrs Vej 38, DK 8220 Brabrand, Aarhus, Denmark, Tel: 0045 8943 5495; Cell: +45 24675 165. External Professor at Lund University Sweden (2008-14). Pls watch our 1 min video on phytase: https://youtu.be/hvQK6XNd9Z0

---

## [Editor Report · Decision Letter 2]

31 Aug 2020

Comparative study of protease hydrolysis reaction demonstrating Normalized Peptide Bond Cleavage Frequency and Protease Substrate Broadness Index

PONE-D-20-14957R2

Dear Dr. Yu,

We’re pleased to inform you that your manuscript has been judged scientifically suitable for publication and will be formally accepted for publication once it meets all outstanding technical requirements.

Kind regards,

Luis Menéndez-Arias, Ph. D.

Academic Editor

PLOS ONE
---

## [Editor Report · Acceptance letter]

4 Sep 2020

PONE-D-20-14957R2 

Comparative study of protease hydrolysis reaction demonstrating Normalized Peptide Bond Cleavage Frequency and Protease Substrate Broadness Index 

Dear Dr. Yu:

I'm pleased to inform you that your manuscript has been deemed suitable for publication in PLOS ONE. Congratulations! Your manuscript is now with our production department. 

Kind regards, 

on behalf of

Dr. Luis Menéndez-Arias 

Academic Editor

PLOS ONE